# Heat current-driven topological spin texture transformations and helical q-vector switching

Fehmi Sami Yasin [1] ✉, Jan Masell [1,2], Kosuke Karube [1], Daisuke Shindo[1], Yasujiro Taguchi [1], Yoshinori Tokura [1,3,4] & Xiuzhen Yu [1] ✉

The use of magnetic states in memory devices has a history dating back decades, and the experimental discovery of magnetic skyrmions and subsequent demonstrations of their control via magnetic fields, heat, and electric/thermal currents have ushered in a new era for spintronics research and development. Recent studies have experimentally discovered the antiskyrmion, the skyrmion's antiparticle, and while several host materials have been identified, control via thermal current remains elusive. In this work, we use thermal current to drive the transformation between skyrmions, antiskyrmions and non-topological bubbles, as well as the switching of helical states in the antiskyrmion-hosting ferromagnet $(Fe_{0.63}Ni_{0.3}Pd_{0.07})_3P$ at room temperature. We discover that a temperature gradient $\nabla T$ drives a transformation from antiskyrmions to non-topological bubbles to skyrmions while under a magnetic field and observe the opposite, unidirectional transformation from skyrmions to antiskyrmions at zero-field, suggesting that the antiskyrmion, more so than the skyrmion, is robustly metastable at zero field.

Information technology has revolutionized the way humans interact with their environment and each other. Integral to modern devices is the use of electrical states in circuits as bits, stored in semiconductor memory. While electron-spin textures are also utilized in modern devices, the recent experimental discovery and real space observation of topologically protected spin textures such as the skyrmion[1,2] (Sky) and antiskyrmion[3] (Asky) have sparked renewed interest in the field of spintronics[4]. In the years since, researchers have identified and grown several new materials hosting magnetic (anti)skyrmions[5]. These include noncentrosymmetric, cubic crystal magnets such as MnSi[1], (Fe,Co)Si[2], FeGe[6] and $(Co_{0.5}Zn_{0.5})_{20-x}Mn_x$[7] in which the crystal chirality allows for the formation of magnetic Skys via the isotropic Dzyaloshinskii-Moriya interaction (DMI), centrosymmetric crystal magnets[8], multilayer thin films in which a heavy metal/magnetic layer combination stabilizes Skys via interfacial DMI[9,10], and Heusler magnets with $D_{2d}$ symmetry or $S_4$ symmetry crystals with uniaxial anisotropy and anisotropic DMI, which host elliptical Skys, non-topological bubbles (NTB) and Askys[3,11-14].

The prototypical stimulus used to manipulate and control electron-spins is electric current, and several studies regarding spin texture dynamics have thus been undertaken including current-driven domain wall motion[15], helical domain modulation wave vector alignment[16], Sky lattice motion[6], and single Sky motion[17]. However, as energy demands increase across the world, interest in technology capable of harvesting the waste heat generated from energy conversion processes such as those in automobiles and thermal power plants has spiked, as an estimated two thirds of the required input energy is converted to heat[18]. As such, the identification of heat-driven processes and materials that enable "green" spintronics is critically important. One such process is the thermal current-driven domain wall[19,20] and Sky motion[21], and recent experiments have confirmed such Sky motion in both insulating[22] and metallic[23,24] magnets. A recent

[1]RIKEN Center for Emergent Matter Science (CEMS), Wako 351-0198, Japan. [2]Institute of Theoretical Solid State Physics, Karlsruhe Institute of Technology (KIT), 76049 Karlsruhe, Germany. [3]Department of Applied Physics, University of Tokyo, Tokyo 113-8656, Japan. [4]Tokyo College, University of Tokyo, Tokyo 113-8656, Japan. ✉e-mail: fehmi.yasin@riken.jp; yu_x@riken.jp

study has also found that externally applied magnetic fields can drive the controlled, reversible topological transformations between Skys and Askys in Asky-hosting magnets[12,13].

In this study, we designed a focused ion beam (FIB)-fabricated device (see Fig. S1 in the Supplementary Information) to apply a range of temperature gradients $\nabla T$ across the room temperature ($T_C \approx 362$ K, see the magnetic state diagram in Fig. S2a), noncentrosymmetric tetragonal crystal $(Fe_{0.63}Ni_{0.3}Pd_{0.07})_3P$ and measure the in-situ response of the native magnetic helices, (anti)skyrmions and NTBs in real space using Lorentz transmission electron microscopy (LTEM). In the following, we summarize the $\nabla T$-driven spin texture transformations observed in the thin plate magnet and then discuss the potential mechanisms for each evidenced by the experimental data.

## Results

### Spin textures inherent to $(Fe_{0.63}Ni_{0.3}Pd_{0.07})_3P$

In $(Fe_{0.63}Ni_{0.3}Pd_{0.07})_3P$, anisotropic DMI stabilizes a ground state magnetic helix with two propagation vectors ($\mathbf{q}$) aligned almost along the [110] and [$\bar{1}$10] crystal axes[12]. Upon application of a temperature gradient $\nabla T = (\partial_x T, 0, 0)$ where the cold side of the device is held at room temperature $T = 295$ K, we observe a transformation from this two $\mathbf{q}$-vector helical state to a single $\mathbf{q}$-vector state ($\mathbf{q} \perp \nabla T$), as drawn schematically in Fig. 1a. By applying a magnetic field perpendicular to the thin plate device, i.e., along the [001] crystal axis, thermodynamically stable Askys, NTBs and Skys can be generated for increasing field values. Even after the magnetic field is turned off, these states may remain in a metastable state embedded in the ground state magnetic helices[13,25]. Figure 1b, c illustrates the $\nabla T$-induced unidirectional topological transformations observed from thermodynamically stable Askys to Skys at non-zero fields, and from metastable Skys to Askys at zero field, where NTBs act as the intermediate state between the topological spin textures. Whereas Skys/Askys exhibit vortex/antivortex real space spin configurations and a topological charge of $N = \frac{1}{4\pi} \int\int d^2\mathbf{r}\, \mathbf{n} \cdot \left( \frac{\partial \mathbf{n}}{\partial x} \times \frac{\partial \mathbf{n}}{\partial y} \right) = \mp 1$, respectively, NTBs show a combined form of half-Sky and half-Asky with one Bloch line pair and hence have zero topological charge ($N = -\frac{1}{2} + \frac{1}{2} = 0$). Using LTEM, we measured the in-plane magnetic induction of each of these states (see the "Methods" section for more details), which reveals the oscillating spins

within the two $\mathbf{q}$-vector helical domain walls (Fig. 1d), the square-like shape of the magnetic Askys (Fig. 1e) indicating the interplay between anisotropic DMI and dipolar interaction, and the elliptical shape of the magnetic Skys (Fig. 1f) characteristic of the anisotropic DMI in this crystal magnet[12].

### $\nabla T$-driven helical q-vector switching

Let us first analyze the zero-field, $\nabla T$-driven transformation of the initial two $\mathbf{q}$-vector helical state, shown in the defocussed LTEM micrograph in Fig. 2a, with $\mathbf{q_1}$ and $\mathbf{q_2}$ almost parallel to the [$\bar{1}$10] and [110] crystal axes, respectively, consistent with previous observations[12]. Upon application of $\nabla T \approx 4.2$ K $\mu m^{-1}$ (see the "Methods" section for a description of how we quantify $\nabla T$), the magnetic contrast disappeared as shown in Fig. 2b. This is because the field of view in Fig. 2a–c is a region of the sample located near the Pt heater wire (inset Fig. 2b), where the temperature for $\nabla T \approx 4.2$ K $\mu m^{-1}$ ranges from $369 - 382$ K. As this temperature range is above $T_C$, the magnetic state in Fig. 2b is paramagnetic. After termination of the heater current, a process where $\nabla T$ decreases linearly to zero over 1 s, magnetic ordering returned in the form of a helical state with $\mathbf{q_2}$ (Fig. 2c).

To explain this magnetic state transformation, we turn to micromagnetic simulations performed with a rectangular geometry to match the experiment (see "Methods" for details). Starting from the two $\mathbf{q}$-vector helical state shown in Fig. 2d, we applied $\nabla T_1^* = 442$ K $\mu m^{-1}$ (Fig. 2e) to drive the temperature in the simulated field of view above $T_C^*$, where $T^*$ is the simulated temperature scaled to match experiment[26]. To simulate the temperature gradient termination procedure described above, we reduced the temperature gradient to $\nabla T_2^* = 4.4$ K $\mu m^{-1}$ (Fig. 2f) and allowed the simulation to run for $10^{-5}$ s. When the helical ordering returned in Fig. 2f, we observed the formation of several magnetic defects including the dislocation defect marked by an arrow, which began to propagate across the sample from cold to hot due to the magnonic torque induced by $\nabla T_2^*$[27]. The motion of these defects effectively combed the spins into the single $\mathbf{q}$-vector helical state seen in Fig. 2g, and once they reached the edge of the sample, the defects annihilated. We then reduced $\nabla T^*$ to zero and relaxed the magnetic system, which remained in a single $\mathbf{q}$-vector helical state.

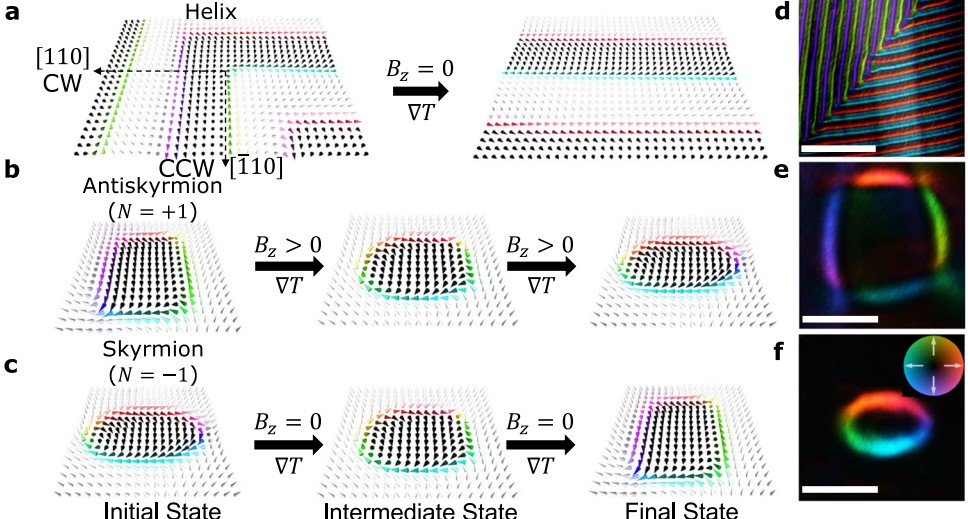

**Fig. 1 | Heat current-driven transformations of magnetic spin textures in $(Fe_{0.63}Ni_{0.3}Pd_{0.07})_3P$.** Schematics of temperature gradient ($\nabla T$)-induced transformations of (**a**) a two modulation $\mathbf{q}$-vector helical state into a single $\mathbf{q}$-vector state at zero field ($B_z = 0$ mT), (**b**) a thermodynamically stable antiskyrmion (Asky, topological charge $N = +1$) into a skyrmion (Sky, $N = -1$) via a non-topological bubble (NTB, $N = 0$) while $B_z > 0$ mT, and (**c**) a metastable Sky into an Asky via a NTB while $B_z = 0$ mT. Measured in-plane magnetic induction maps of (**d**) helical stripes at zero field with CW and CCW helicities aligned with $\mathbf{q}$-vectors parallel to the [110] and [$\bar{1}$10] crystal axes, respectively, (**e**) an Asky with topological charge $N = +1$, and (**f**) a Sky with topological charge $N = -1$, all acquired in the FIB-fabricated $(Fe_{0.63}Ni_{0.3}Pd_{0.07})_3P$ heating devices used in this study. Scale bars 1 μm in (**d**) and 0.1 μm in (**e**, **f**).

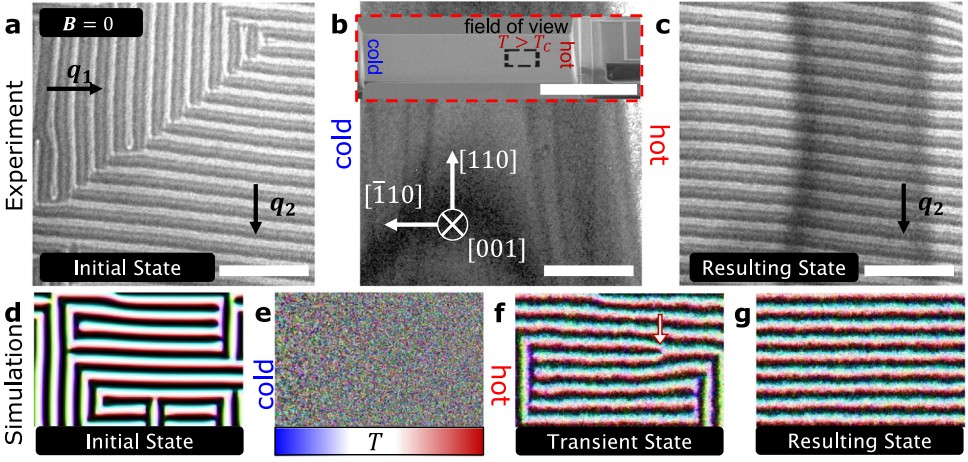

**Fig. 2 | Heat current-driven ordering of a double q-vector helical state to a single q-vector alignment at zero field.** LTEM micrographs of the (**a**) initial double **q**-vector helical state at 295 K, (**b**) paramagnetic state during the application of $\nabla T \approx 4.2$ K $\mu m^{-1}$ which causes an increase in the base temperature to between 369 K and 382 K in the field of view and (**c**) resulting single **q**-vector helical state upon termination of the heater current and cooling of the field of view to a uniform 295 K in $\approx 1$ s. The inset outlined by dashed red lines in (**b**) is a scanning electron micrograph taken with a 52° sample tilt and indicates the field of view ($3.2 \times 3.2$ $\mu m^2$, outlined by dashed black lines) of the experiment (**a**–**c**), in which

$T > T_C$. Micromagnetic simulations of $\nabla T$-driven ordering of the magnetic domains from the (**d**) initial double **q**-vector helical state to a (**e**) paramagnetic state during the application of scaled temperature gradient $\nabla T_1^* = 442$ K $\mu m^{-1}$, (**f**) partially aligned double **q**-vector helical state during the application of $\nabla T_2^* = 4.4$ K $\mu m^{-1}$ and (**g**) resulting single **q**-vector helical state after applying $\nabla T_2^*$ for a sufficiently long time. A magnetic defect in the helical state shown in (**f**) is indicated by an arrow. The heater is located on the right-hand-side of each image, while the cold bath is on the left-hand-side. Scale bars are 1 $\mu m$ in (**a**–**c**) and 10 $\mu m$ in the inset in (**b**).

Notably, magnetic defect motion has been utilized for combing magnetic helices before by an electric current in FeGe, a cubic crystal with broken inversion symmetry[16]. The helices in $(Fe_{0.63}Ni_{0.3}Pd_{0.07})_3P$ are strongly pinned, resulting in magnetic defect motion perpendicular to the **q**-vector's direction and laterally confined within the helices, similar to the recent electric current-driven defect and Sky motion in FeGe[28]. We confirmed the cold-to-hot $\nabla T$-driven motion of these magnetic defects experimentally in Fig. S3. Additionally, we performed an experiment in which we uniformly heated the two **q**-vector helical state above $T_C$ and then cooled to room temperature, resulting in the return of the two **q**-vector helical state as shown in Fig. S4. This evidence, corroborated by micromagnetic simulations, suggests that $\nabla T$-driven motion of magnetic defects drives the helical transformation.

### $\nabla T$-driven topological transformation of Asky to Skys at finite field

Now let us turn our gaze toward the topological spin textures inherent to $(Fe_{0.63}Ni_{0.3}Pd_{0.07})_3P$. Applying an external magnetic field $B_z \parallel [001] = 439$ mT drives the transformation from a helical state to Askys mixed with NTBs embedded in a field-polarized background, as shown in Fig. S5a. While holding the magnetic field constant, we applied $\nabla T = 9.4 \times 10^{-2}$ K $\mu m^{-1}$ across the thin plate magnet and observed the topological transformation from Askys to NTBs shown in Fig. S5b. The number of Asky to NTB transformations further increased when we increased the temperature gradient to $\nabla T = 3.3 \times 10^{-1}$ K $\mu m^{-1}$ (Fig. S5c), and now an NTB further transformed into a Sky. Selected transformations are indicated by arrows, with red, green, and blue arrows indicating Asky, NTB and Sky, respectively. In this unidirectional transition from Asky to NTB to Sky, the latter spin textures stabilized on the hot side of the field of view, suggesting that although $\nabla T \neq 0$, the transformation was thermally driven. Indeed, a transition from Asky to NTBs to Sky is observed when the thin plates are heated uniformly, consistent with the previously reported $(Fe_{0.63}Ni_{0.3}Pd_{0.07})_3P$ magnetic state diagram[12].

To confirm this, we performed numerical calculations of the energy of an isolated Asky and Sky in a field-polarized background held at a constant magnetic field. We varied the saturation magnetization

$(M_S)$ together with all other micromagnetic parameters which depend on $M_S$ which is an effective description of the system at finite temperatures as the magnetization is a monotonously decreasing function of the temperature. We plot the resulting energy versus magnetization in Fig. S5d. The inset panels show the simulated real-space images of the relaxed Asky and Sky at various indicated magnetization values. Several notable features emerged from these calculations, such as the decrease in spin texture size as magnetization decreases (effectively equivalent to increasing $T$), which is consistent with experiment. For the lowest magnetization $M_S \approx 4.25 \times 10^5$ A $m^{-1}$ (largest temperature) the Sky and Asky collapse and cease to exist even as metastable excitations. Above this critical magnetization, the energy of both the Sky and Asky is first positive, meaning that the defect-free field-polarized state is the ground state, but they eventually turn negative above some value of the magnetization which marks a phase transition to a Sky lattice or gas. Interestingly, we find that $\Delta E = E_{Sky} - E_{Asky} < 0$ in a magnetic polarized background for every $M_S$ value considered here and so as $T$ increases (decreasing $M_S$) a transformation from Asky toward Sky should be expected. Another feature is the elongation of Askys into rectangular geometries (not shown here), which decreases their energy compared to the Skys. Such elongated Askys are also present in experiment, as we show later. However, while Skys are clearly favorable over Askys in this high temperature limit, we cannot extrapolate this data to the low temperature limit as the embedding background is very different in the two cases which will drastically alter the results.

### $\nabla T$-driven topological transformation of Skys to Askys at zero field

We observed a more puzzling phenomenon when we applied $\nabla T$ to metastable Skys, generated by first applying a magnetic field sufficient to stabilize a thermodynamically stable Sky lattice phase and then turning the field off, as shown on the left-hand-side (LHS) of Fig. 3a. The resulting metastable Skys (with some NTBs and Askys) are shown in Fig. 3a–c shows the spin textures after applications of $\nabla T = 2.0$ K $\mu m^{-1}$ and $\nabla T = 5.2$ K $\mu m^{-1}$, respectively, with arrows indicating the locations of selected spin textures as they transformed unidirectionally from Skys (blue, Fig. 3a) to NTBs (green, Fig. 3b) to

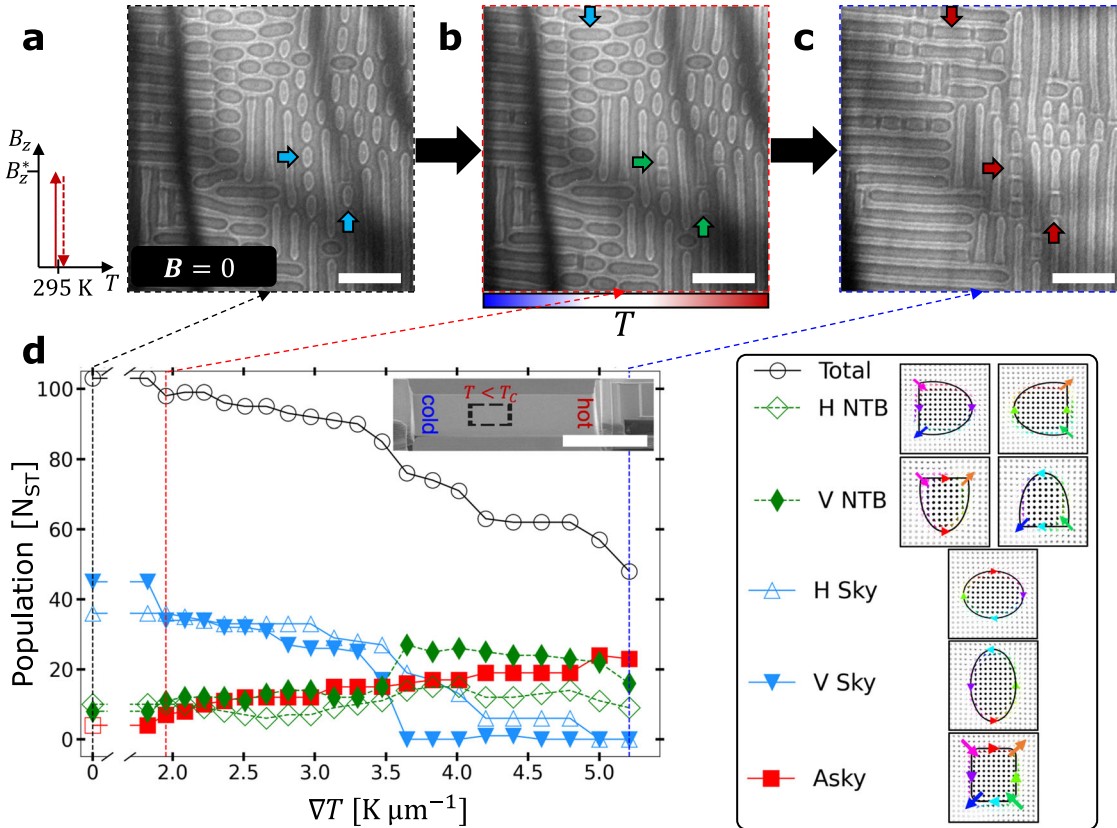

**Fig. 3 | Unidirectional transformation of topological spin textures at zero field via heat current. a** LTEM micrograph of the initial magnetic state composed of metastable spin textures stabilized by applying $B_z$ to form a Sky lattice state and then reducing the field to zero at 295 K (schematic on LHS of **a**), after which some skyrmions remained stable. LTEM micrographs of the zero-field magnetic states taken after heater current applications of (**b**) $\nabla T = 2.0$ K μm⁻¹ (314 K < $T_{FOV}$ < 323 K, where $T_{FOV}$ is the temperature in the field of view) and (**c**) $\nabla T = 5.2$ K μm⁻¹ (343 K < $T_{FOV}$ < 368 K) lasting 15 s (except for those values noted in the "Methods"). Selected spin textures are marked as they transform from Sky (blue arrows) to NTBs (green arrows) to Askys (red arrows). **d** Spin texture population ($N_{ST}$ is the combined number of antiskyrmions Asky, nontopological bubbles NTB, and skyrmions Sky) versus $\nabla T$ of the full 7.3 × 5.5 μm² field of view in which the 4.6 × 4.6 μm² area shown in (**a**–**c**) is contained. Five types of spin textures are identified, including Askys (red square markers), horizontal Skys ("H Sky" with open blue triangle markers), vertical skyrmions ("V Sky" with blue, upside-down triangle markers), horizontal non-topological bubbles ("H NTB" with open green equal-length-diagonal rhombus markers), and vertical non-topological bubbles ("V NTB" with green rhombus markers). The total number of spin textures is plotted by a solid black line with open circular markers. The inset panel shows the field of view of the experiment within the (Fe₀.₆₃Ni₀.₃Pd₀.₀₇)₃P device, and the right-hand-side legend shows schematics of the in-plane magnetization for each spin texture measured in (**a**–**d**). The scale bar in (**a**–**c**) is 1 μm and 10 μm in the inset panel of (**d**).

Askys (red, Fig. 3c). A summary of the spin texture population as a function of increasing $\nabla T$ within the observed 7.3 × 5.5 μm² field of view is shown in Fig. 3d. The Asky and Sky populations monotonically increase and decrease, respectively, with increasing $\nabla T$, whereas the NTB population increases until $\nabla T \approx 3.5$ K μm⁻¹, after which the NTB population decreases. We distinguished seven types of spin textures, shown schematically on the RHS of Fig. 3d, including the Asky (open red square marker), elliptical Sky with major axis oriented horizontally (open blue triangle marker) or vertically (filled blue upside-down triangle marker), and NTB oriented either horizontally with Bloch lines located on the left or right side of the NTB (open green diamond marker) or vertically with Bloch lines on the top or bottom side (filled green rhombus marker). We note that even though there are cases of elongated Askys, we bin all Askys together. The total number of these seven spin textures are plotted using open black circle markers.

We emphasize that these spin texture populations were measured at zero field, where the helical state is the lowest energy state at all temperatures as reported in the magnetic state diagram[12]. Therefore, the $\nabla T$-driven unidirectional topological transformation we observed here suggests a warped energy landscape in which the presence of Bloch line pairs at the ends of spin textures is preferred. This manifested itself not only as the monotonic increase of the Asky population

with increasing temperature gradient, but also as the reduction of the Sky population to zero. The NTB population, meanwhile, seemingly acts as an intermediate state between the two oppositely charged topological spin textures. As shown in Supplementary Movie 1, the Sky first passes through an NTB state via the $\nabla T$-driven creation of a Bloch line pair. Intriguingly, the data shows a large vertical Sky population collapse and corresponding increase in vertical NTBs between $|\nabla T| = 3.4$ and 3.6 K μm⁻¹. This may be explained by the interaction of neighboring spin textures via the interplay of exchange interaction and dipole-dipole interaction. Once a single Sky within a row of Skys (as seen in the upper RHS of Fig. 3a, b, for example) undergoes a $\nabla T$-driven transformation to an NTB via bottom-end Bloch line pair creation, the spins composing the Bloch line pair may provide sufficient torque to the neighboring spins via the exchange interaction and neighboring Skys' spins via the long-range dipole-dipole interaction to form a chain of transformations, resulting in the alignment of Bloch lines within a row of spin textures. On the other hand, the horizontal Sky population decreases steadily to zero without such a sharp change, indicating that the orientation (vertical or horizontal) of the spin textures may play an important role. We note that we omit data points for $\nabla T > 5.2$ K μm⁻¹ because the temperature on the hot side of the field of view rose above $T_C$, leading to the temperature-induced annihilation of all spin textures in these regions.

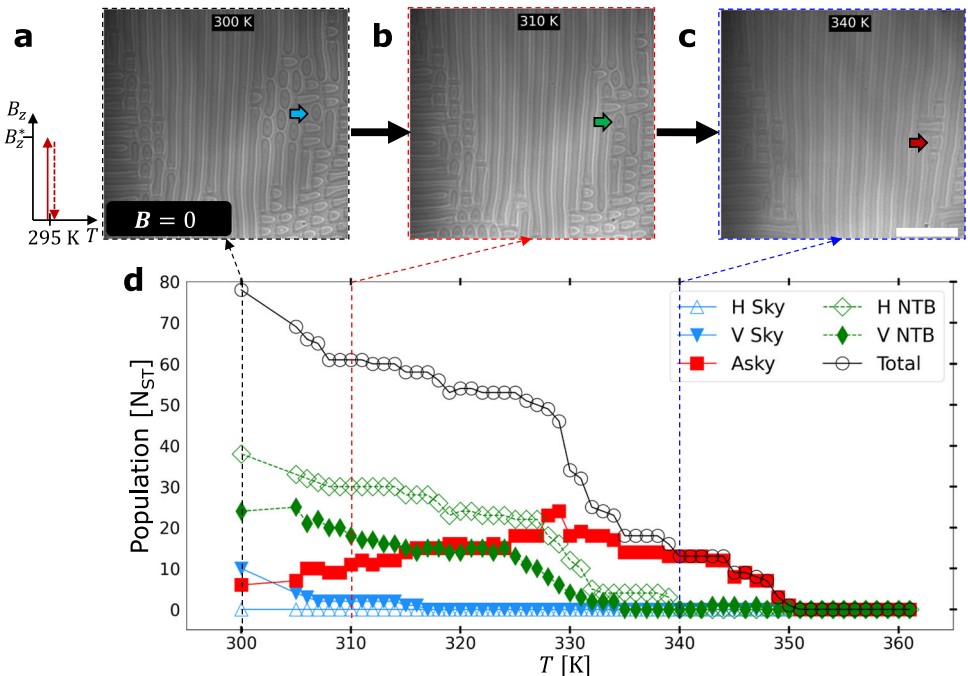

**Fig. 4 | Unidirectional transformation of topological spin textures at zero field via uniform heating. a** LTEM micrographs of metastable spin textures stabilized by applying $B_z$ to stabilize a Sky lattice state and then reducing the field to zero at $T = 295$ K (spin texture creation process outlined in the schematic on LHS of **a**), after which some Skys remained stable and the temperature was increased to 300 K before acquisition of (**a**). LTEM micrographs of the zero-field magnetic states taken at the further elevated temperatures of (**b**) $T = 310$ k and (**c**) $T = 340$ k, respectively. A selected spin texture is tracked from panel to panel and marked in each panel as it transforms from (**a**) Sky (blue arrow) to (**b**) NTB (green arrow) and finally to (**c**) Asky (red arrows). **d** Spin texture population versus $T$ of the

4.0 × 3.8 μm² field of view. Five types of spin textures are identified (drawn schematically in Fig. 3), including Askys (open red square markers), horizontal Skys (labeled "H Sky" with open blue triangle markers), vertical Skys (labeled "V Sky" with blue, upside-down triangle markers), horizontal NTBs (labeled "H NTB" with open green equal-length-diagonal rhombus markers), and vertical NTBs (labeled "V NTB" with green rhombus markers). Asky, H Sky and V Sky are plotted with solid lines, while H NTB and V NTB are plotted with dashed lines. The total number of spin textures is plotted by a solid black line with open circular markers. Scale bar in (**c**) is 1 μm.

To further clarify the role of temperature in $\nabla T$-driven transformations, we performed a uniform heating experiment in a thin plate of similar geometry (see Fig. S2b) in which we increased the temperature uniformly and held it constant for 30–60 s to allow the spin textures to stabilize before acquiring a real space image and increasing the temperature further. We used the same sample geometry as in Fig. 3 so as to contribute any differences in results to the different driving force used, i.e., uniform heating. The results are summarized in Fig. 4. The initial metastable state shown in Fig. 4a was prepared in the same way as in Fig. 3a, and consists of the same mixture of metastable Skys, NTBs, and Askys, although with slightly different initial populations. Figure 4b, c shows the real space images at $T = 310$ K and $T = 340$ K, respectively. The spin texture population is plotted in Fig. 4d with the same markers and plotting as in Fig. 3. Just as in the $\nabla T$ case, the Asky population unidirectionally increases accompanied by decreases in all other spin texture populations until $T = 329$ K, after which the Asky population begins to decrease. This suggests that the $\nabla T$-driven transformations shown in Fig. 3 at zero field may be driven in fact by thermal fluctuations, rather than via magnonic or spin transfer torque. In other words, $\nabla T$ is a sufficient condition for this topological transformation to occur. Importantly though, the temperature range in the field of view at the highest $\nabla T$ value (Fig. 3c) is 343 K $< T_{FOV} <$ 368 K, and at this value, we didn't observe a significant decrease in antiskyrmion population. In contrast, the antiskyrmion population begins to decrease at $T = 330$ K during the uniform heating experiment (Fig. 4d), suggesting that while the two experiments are qualitatively similar, they differ quantitatively, and the temperature gradient contributes to the antiskyrmions' robust metastability at zero field. Although Askys

remain metastable for higher temperatures than any other spin texture, its population eventually reduces to zero at $T = 351$ K, and the thin plate is then populated by the helical state. The full results of this experiment can be seen in Supplementary Movie 2. We performed micromagnetic simulations at zero field for Skys, Askys and helical states while varying temperature and found that the Skys and Askys would relax into the helical state within our model, which agrees with the previously published literature[12]. This contradicts the experimental findings shown here and suggests that more extensive investigations and perhaps new theory are necessary to explain this delicate process.

Let us outline an argument explaining this result qualitatively. To change the topology of the magnetic texture, one needs to introduce a topological defect called Bloch point or anti-Bloch point (or magnetic hedgehog), a singularity in the magnetization around which the magnetization points in all possible directions. At this topological defect the magnetization changes rapidly in space which costs a lot of exchange energy. Accordingly, we can expect that the energy of a Bloch point inside the closed domain wall that constitutes an (anti)skyrmion/bubble is lower than its energy in the polarized background between textures as the magnetization inside the domain wall is already twisted. Thus, a Bloch point is more likely to emerge in the textures' domain walls, rendering topological transitions between isolated defects such as in the transition from Sky to NTB to Asky observed here, rather than the transition to the helical state which requires the fusion of neighboring defects via a Bloch point in the polarized background space between the texture and either its neighbor or the helical background.

## Discussion

These results illustrate the diversity of responses spin textures display while under a temperature gradient. Of these, heat current-driven helical switching stands out as particularly useful for memory applications[29]. Furthermore, the controlled transformation of topological magnetic states beyond room temperature is critical for spintronics devices, and these results uncover the $\nabla T$-induced transformations in Asky-hosting thin magnets and hint at their mechanisms. How these real space observations relate to thermoelectric effects is still unknown, but the experiments presented here pave the way for simultaneous measurement of both.

## Methods

### LTEM imaging and temperature-gradient application

We performed the defocused LTEM experiments in a transmission electron microscope (JEM-2100F, JEOL) equipped with a double-tilt electrical holder (Protochips: Fusion select). The sample holder was powered using a Keithley 2450 SMU. We applied external magnetic fields to the (001) $(Fe_{0.63}Ni_{0.3}Pd_{0.07})_3P$ devices by exciting the objective lens current of the JEM-2100F, which generates a field parallel to the incident electron beam. We also performed LTEM uniform heating experiments in the Thermo Fisher Scientific Talos F200x scanning transmission electron microscope (STEM) using the Thermo Fisher Scientific NanoEx i/v MEMS device TEM heating holder. LTEM images are the summation of the electron-specimen interaction through the thickness projected onto a 2D detector image plane. While holding the defocus constant at $|\Delta f| < 1$ mm and the externally applied magnetic field at $\mu_o H_{ext} = 439$ mT or $\mu_o H_{ext} = 0$, we varied the heater wire current and applied current pulses lasting 5 s for $\nabla T \leq 2.1$ K $\mu m^{-1}$, 60 s for $\nabla T = 3.8$ K $\mu m^{-1}$, and 15 s for all other $\nabla T$ values. When two LTEM images of a magnetic spin texture are acquired at over- and under-defocus values, the in-plane magnetic inductance of the sample may be reconstructed using the transport-of-intensity equation, assuming a flat electrostatic phase distribution (uniform thickness).

### Micromagnetic simulations

We performed micromagnetic simulations based on the (stochastic) Landau-Lifshitz-Gilbert (sLLG) equation using the MuMax3 software in two distinct fashions for the data shown in Figs. 2 and S3, respectively[30].

First, for the simulations of dynamical **q**-vector switching in Fig. 2, the size of the magnet was $L_x \times L_y \times L_z = 4096 \times 2048 \times 128$ nm³ discretized into $256 \times 128 \times 8$ cuboids, grouped into 256 slices of $1 \times 128 \times 8$ cuboids. The 256 slices were held at different temperatures to achieve $\nabla T^* \neq 0$ across the magnet, and we set open boundary conditions. The temperature in this case was modeled via a fluctuating magnetic field as implemented in Mumax3. We can express the Curie temperature as $T_C^* \approx 2Aa_L/k_B$,[26] where $A$ is the exchange stiffness, $a_L$ is the lattice constant (simulated mesh voxel size) and $k_B$ is Boltzmann's constant. We set the temperature on the cold side of the simulation in Fig. 2 to be $T_{cold}^* = 0.95\ T_C$, with $\nabla T_1^* = 5\ T_C^*/L_x$ (Fig. 2e) and $\nabla T_2^* = 0.05\ T_C^*/L_x$ (Fig. 2e). The material parameters were chosen according to ref. 12 at $T = 300$ K: saturated magnetization $M_S = 4.17 \times 10^5$ A m⁻¹; exchange stiffness $A = 8.1 \times 10^{-12}$ J m⁻¹; anisotropic DMI constant $D_{ani} = 2.0 \times 10^{-4}$ J m⁻² and uniaxial anisotropy constant $K_u = 3.1 \times 10^4$ J m⁻³ along the $z$ axis. The Gilbert damping constant was set to $\alpha = 0.03$. To achieve anisotropic DMI, we added custom energy density and effective field terms using the MuMax3 built-in functions. The field term is of the form $\mathbf{B}_{D_{ani}} = 2D_{ani}(\partial_y n_z \hat{x} + \partial_x n_z \hat{y} - (\partial_x n_y + \partial_y n_x)\hat{z})$, where $n_i$ is the $i$th component of the magnetization, and the energy density expression is defined as $E_{D_{ani}} = -\frac{1}{2}\mathbf{n} \cdot \mathbf{B}_{D_{ani}}$.[13,14,30]

In turn, for the static simulations of the energy in Fig. S5 we applied a simple local energy minimum search algorithm at $T = 0$ K, starting from a skyrmion, antiskyrmion, or simply polarized state, respectively. The system size is $1000 \times 1000 \times 200$ nm³ discretized into $256 \times 256 \times 64$ cuboids with periodic boundary conditions in the x-y-plane and using a modified version of MuMax3 where gradients are discretized via fourth order finite differences, similar to previous works[12,31]. The parameters are taken from ref. 12 at $T = 2$ K, i.e., $M_s = 6.65 \times 10^5$ A m⁻¹, $A = 20 \times 10^{-12}$ J m⁻¹, $D_{ani} = 4.0 \times 10^{-4}$ J m⁻², and $K_u = 1.12 \times 10^5$ J m⁻³. For different values of the saturation magnetization (temperature), we rescale the micromagnetic parameters. We scale $A \propto M_S^2$, $D \propto M_S^2$, and $K_u \propto M_S^3$.[32,33]

In both simulations, the $x$ and $y$ axes correspond to the experimental [110] and [1$\bar{1}$0] crystal axes, respectively, with a righthanded helix stabilized in x-direction and a lefthanded helix in the y-direction.

### Temperature gradient approximation

To approximate $\nabla T$ across the thin plate device, we applied a current $I_{heater}$ to the heater wire until the magnetic contrast in LTEM began to disappear at the hot end (nearest to the wire) of the thin plate. This loss of contrast occurred along a clear line perpendicular to the temperature gradient direction and is the result of the magnet heating above $T_C$ and thus entering the paramagnetic regime. As expected, this $T_C$ line propagates further toward the cold end of the sample as $I_{heater}$ increases and may be used as a thermometer to measure the location of $T_C$ as a function of $I_{heater}$. After measuring the $T_C(I_{heater})$ curve from several LTEM micrographs (shown in Figs. S6 and S7 for the two devices used here) at various $I_{heater}$ values, we calculated $\nabla T(I_{heater})$ assuming that $T_{cold\ bath} = 295$ K.

### Sample preparation

Single-crystalline bulk samples of $(Fe_{0.63}Ni_{0.3}Pd_{0.07})_3P$ were synthesized by a self-flux method from pure Fe, Ni and Pd metals and red phosphorous sealed in an evacuated quartz tube. The target phase of tetragonal $M_3P$ was isolated from the ingot. Phase purity of the $M_3P$ structure was confirmed by powder X-ray diffraction with Cu Kα radiation[12]. Crystal orientations were checked by an X-ray Laue diffraction method. Chemical compositions were examined by a scanning electron microscope equipped with an energy dispersive X-ray analyzer[12].

### In-plane magnetic induction field maps

The in-plane magnetic induction field distribution was obtained by the transport-of-intensity equation (TIE) analysis[34] of the over-focus and under-focus LTEM images and displayed by color mapping.

### Classification of spin texture populations

In the zero-field LTEM micrographs analyzed for Figs. 3 and 4, we classified the spin textures as either Sky, Asky, NTB or as part of the helical background. The spin textures were classified as helices if they extended beyond the field of view of observation. Therefore, it was possible to have some "long-length" Skys, Askys and NTBs.

### Estimation of standard deviation of $\nabla T$

If we assume we can only measure the distance of the location of $T_C$ to the cold bath($x_{T_C}$, distance of the location of loss of magnetic contrast in Figs. S6 and S7 to the cold bath) to within one helical period, we can define the uncertainty of each measurement to be $\sigma_{x_{T_C}} = 200$ nm. The temperature gradient may be measured from each image as described above, $\nabla T = \frac{T_C - T_{RT}}{x_{T_C}}$. Therefore, we may calculate the uncertainty in $\nabla T$ to be $\sigma_{\nabla T}^2 = \sigma_{x_{T_C}}^2 (\frac{\partial \nabla T}{\partial x_{T_C}})^2 = \sigma_{x_{T_C}}^2 (\frac{T_C - T_{RT}}{-x_{T_C}^2})^2$.

## Data availability

The data that support the plots within this article are available at https://doi.org/10.6084/m9.figshare.23857581.v1.

## Code availability

The code used to generate the micromagnetic simulations should be reproducible from the descriptions provided in the "Micromagnetic simulations" section. Otherwise, the code is available from the corresponding author upon reasonable request.

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

## Acknowledgements

We are very grateful to Tomoko Kikitsu (Materials Characterization Support Team in the RIKEN Center for Emergent Matter Science) for technical support on the TEM (JEM-2100F) and Naoto Nagaosa for helpful discussions. Y. Tokura acknowledges the support of Japan Science and Technology Agency (JST) CREST program (Grant Number JPMJCR1874). X.Y. acknowledges the support of Grants-In-Aid for Scientific Research (A) (Grant No. 19H00660) from JSPS and JST-CREST program (Grant No. JPMJCR20T1). J.M. was supported by the Alexander von Humboldt Foundation as a Feodor Lynen Return Fellow.

## Author contributions

F.S.Y., Y. Tokura and X.Y. jointly conceived the project with input from D.S. K.K. and Y. Taguchi synthesized the bulk crystals. F.S.Y. fabricated the FIB heating devices, performed LTEM measurements and all data analyses. J.M. and F.S.Y. jointly contributed the theoretical model for helical q-vector ordering and spin texture transformation. F.S.Y. wrote the manuscript, and the results were discussed and interpreted by all the authors.

## Competing interests

The authors declare no competing interests.
