## [Peer Review File · Nature Communications]

Reviewers' Comments:

Reviewer #1:

Remarks to the Author:

Yasin et al. performed an experimental investigation of antiskyrmion-hosting ferromagnet ($\text{Fe}_{0.63}\text{Ni}_{0.3}\text{Pd}_{0.07}$)₃P under the influence of thermal gradients. They showed that the application of a thermal gradient drives the transformation of antiskyrmions into non-topological solitons and then skyrmions. The opposite transformations occur starting from a skyrmion lattice. Micromagnetic simulations are also performed to confirm the experimental results. The paper is well-written, well-organized and easy to read. The topic of combination of thermal gradients and solitons is very timely, and the results achieved are interesting. However, I am not convinced that the results are due to the application of the thermal gradients, as also the authors state in the papers. Therefore, I cannot recommend the publication unless the authors unequivocally prove the key role of the thermal gradients, by addressing the following comments:

1. Paragraph DT-driven helical q-vector switching

The results of Fig. 2 should be confirmed both in the experimental system and simulations. Authors should prove that that transformation does not occur with small increase of a uniform temperature but only under a thermal gradient.

2. Paragraph DT-driven topological transformation of Asky to Skys at finite field.

Authors say: "Indeed, a transition from Asky to NTBs to Sky is observed when the thin plates are heated uniformly consistent with the previously reported ($\text{Fe}_{0.63}\text{Ni}_{0.3}\text{Pd}_{0.07}$)₃P magnetic state diagram¹²." Then, what is the significance of the thermal gradient? The key role of the thermal gradient should be clearly evident, otherwise the achievements are only incremental results of previous works cited by the authors.

3. Paragraph DT-driven topological transformation of Asky to Skys at finite field.

I find this transformation quite obvious considering the field protocol used to stabilize skyrmions at zero field. Skyrmions remain trapped in the high-field metastable state due to the sudden field removal. The following application of larger temperature, restores the ground state made by helices.

Again, which is the role of the thermal gradient compared to uniform increased temperature?

Indeed, authors commenting the results of Fig. 5 say that is not the consequence of thermal gradient "This suggests that the DT-driven transformations shown in Fig. 4 at zero field may be driven in fact by thermal fluctuations, rather than via magnonic or spin transfer torque"

4. Also, I did not understand if the transformations in Fig. 5 start from the same initial configuration of Fig. 4. Fig. 5b is not the same as Fig. 5a, while, to prove the authors result, the initial state must be the same and check how it evolves with a uniform increasing temperature.

5. Why no skyrmion/antiskyrmion motion is observed? Such as in Phys. Rev. B 106, 024415, July 2022.

6. Also, apart from the phase transitions - the same group has shown similar phase transitions in a previous work Ref. 12 - , which is the importance of these results for applications? Motivation, significant advancements in the field as well as a perspective are only slightly mentioned or missing.

Other comments:

- The motivation should be expanded for a broad audience. Why is it important to use thermal manipulation of solitons over other methodologies, such as electrical?

- Figure 2b is unclear. Where is the Pt wire? What is the white rectangle? Which are the dimensions of the sample in the inset and in the main panel?

- Which is the temperature dependence of M_s in the simulations? It should be clearly added to Methods.

- Since many combinations of field and temperature are considered in the main text, I suggest to

move the phase diagram of Fig. S2a to the main text and make it clearer. Authors can mark the phase diagram points where the images are taken. Skyrmion lattice should be also highlighted. FM and PM regions should have different colors.

- Fig. 4a looks more like a mixed phase of skyrmions and helices rather than a skyrmion lattice.
- In general, the experimental figures lack resolution quality, especially when compared with the images in Ref. 12 from the same group. Is it possible to increase images definition?

Reviewer #2:

Remarks to the Author:

In the manuscript "Heat current-driven topological spin texture transformations and helical q-vector switching", the authors study how heat gradient could trigger the switching between different topological magnetic nanostructures (which are antiskyrmions, non-topological bubbles, and skyrmions) in the material system with S4 point group crystal symmetry. The material system with S4 (as well as D2d) symmetry is a novel new material system with anisotropy DMI interaction, thus enabling magnetic nanostructure with different topologies to be stabilized, which is a novel platform for the non-volatile memory device. Then the question to the community is how to control switching between these metastable spin structures. In their previous work (ref. 12, 13), the authors show that with the in-plane field induced by tilting the sample, such switching could be achieved. In this work, the authors further show that using heat gradient could also achieve such switching. This should be the first experiment attempt that uses heat gradient to control spin topology switching in the S4/D2d-based material system, which is clear progress in skyrmion manipulation.

In the manuscript, the following experiment results support the author's conclusion:

- A. After applying the heat gradient, the original 2-q helix state becomes a single-q helix state, and the single helix state direction corresponds to the heat gradient direction, as shown in Fig. 2a-c.
- B. With the heat gradient, magnetic defects move along the heat gradient direction, from cold to hot, as shown in Extended Data Fig. 1.
- C. With the field applied, under the suitable heat gradient, antiskyrmion switches to the non-topological bubble and finally into skyrmion, as shown in Fig. 3a-c.
- D. Without field applied, under the suitable heat gradient, skyrmion switches to the non-topological bubble and finally into antiskyrmion, as shown in Fig. 4a-c.
- E. The difference between Fig. 4d and Fig. 5d shows that the effect is not induced purely by uniform heat that heat gradient also contributes.

The underlying physics, especially how the heat gradient generated torque achieves the switching, could be very interesting and complicated. This work, on the one hand, further confirms the metastability of the spin texture D2d/S4 system. On the other hand, demonstrates the possibility of using heat current to control skyrmion topology, which might encourage more heat gradient-driven skyrmion dynamic studies. So, I recommend this manuscript be published in Nature Communications with some possible modifications, as commented below:

#1. There are two temperatures of relevance: temperature on the cold side of the device and temperature difference across the device. As shown by Extended Data Fig. 2, the authors can nicely calibrate the temperature. Could the authors state the two temperatures of each measurement clearly? For example, in line 60, the cold side is 295K; In line 82, the temperature range is 369K-382K, so I guess the cold side is 369K; In line 88-90, the simulated temperature gradient is given, but the temperature at the cold edge seems not given.

#2. Due to the similarity in sample composition, I assume the sample used in this paper is the same as the one used in the author's previous work (Ref. 12). In Fig. 1h of Ref. 12, the Ms have dramatical dependence of temperature when the temperature is above ~350K, which is the temperature range studied in this work. So, is the heat gradient-induced switching of skyrmion topology only occur just below Tc?

#3. In line 91, the simulated temperature gradient is chosen to be 4.9K/um, where helical order

returns. In line 80, the experiment temperature gradient is 4.2K/um (which is lower than 4.9K/um), where the magnetic contrast disappears due to the sample temperature going above T_c . The absolute temperature used in the micromagnetic simulation usually does not have a one-to-one correspondence to the real experiment value. For example, its effect also depends on the simulated cell size chosen. The authors may want to modify this part to avoid confusion.

#4. There is a difference in DMI between the S4 and the D2d system. The S4 point group has generators of $(-x,-y,z)$ and $(y,-x,-z)$. The D2d point group has one more generator of $(-x,y,-z)$. For the D2d point group, the DMI Hamiltonian is $H_{D2d}=D(My \partial Mz/\partial x - Mz \partial My/\partial x) - D(Mz \partial Mx/\partial y - Mx \partial Mz/\partial y)$; For the S4 point group, the symmetry is lower than that in D2d and the DMI Hamiltonian is: $H_{S4}=D1(My \partial Mz/\partial x - Mz \partial My/\partial x) - D1(Mz \partial Mx/\partial y - Mx \partial Mz/\partial y) + D2(My \partial Mz/\partial y - Mz \partial My/\partial y) + D2(Mz \partial Mx/\partial x - Mx \partial Mz/\partial x)$. By changing the coordinate system that rotates along the Z-axis for a certain angle regarding D1 and D2, H_{S4} could transform into the form of H_{D2d} . This means that due to the lower symmetry in the S4 system, the DMI preferred Bloch direction is not exactly along [100] but with a certain angle related to D1 and D2. This angle can be clearly observed in Fig. 2a. The authors should modify the corresponding parts.

#5. In Fig. 2b, the temperature is 369K-382K which is larger than $T_c=362K$. Is heating the device above T_c a must for switching from the 2-q state to the 1-q state? Since Extended Data Fig. 1 shows that defect could move along the heat gradient, the switching seems could be achieved below T_c already.

#6. The bending contours are different between Fig. 2a and Fig. 2c and between Extended Data Fig. 1 a-d. Why is that? Maybe comes from temperature difference?

#7. In line 103, the authors confirm the cold-to-hot heat gradient-driven motion of magnetic defects and show it in Extended Data Fig. 1. What's the velocity of this motion? And is the velocity dependent on the heat gradient amplitude? Even more interesting is that will antiskyrmion/non-topological bubble/skyrmion also have similar heat gradient-driven motion? And since these topological magnetic nanostructures are anisotropic, will heat gradient be applied along different crystalline directions to trigger different dynamics? This question may go beyond the scope of this paper since heat gradient measurement in LTEM itself is already not easy.

#8. The similarity and difference between Fig. 4d and Fig. 5d show that both uniform heat and heat gradients contribute to the final switching behavior. It would be nice if the authors could discuss the underlying physics mechanism.

#9. Under zero field, the helix should be the ground state. In Fig. 4 and Fig. 5, when analyzing the skyrmion/non-topological bubble/antiskyrmion number, are the long-length magnetic topological structure counts as 'skyrmion' etc. or 'short helix'?

#10. Is the heat gradient effect in this S4 material system similar to the other system reported before, such as Bloch skyrmion in B20?

Below are a few minor points:

#1. Ref. 25 is cited in line 90, which is the book that seems not so related to the temperature discussed there.

#2. In Fig. 1d, the two helix directions have an angle away from 90 degrees (different than that in Fig. 2a). Is this due to local pinning?

#3. In supplementary video 2, at above 359K, the helix changes its direction. Is it due to heat gradient-induced alignment?

#4. In lines 45-48, the author claims their previous studies used a combination of external magnetic field and heat to transform skyrmion and reference Ref. 12-13. But it seems only the external field is applied. Or, by heat, do the authors mean controlling the temperature?

#5. In previous work, the sample is $\text{Fe}_{1.9}\text{Ni}_{0.9}\text{Pd}_{0.2}\text{P}$. In this work, the authors use $(\text{Fe}_{0.63}\text{Ni}_{0.3}\text{Pd}_{0.07})_3\text{P}$. Are they the same?

#6. In line 57 and line 100 etc., the authors use the word 'pins'. 'Pins' may indicate the external artifact-induced effect. Here, the Bloch-type chirality is preferred by the intrinsic DMI energy term. The author may want to change the word.

#7. In lines 135-136, the author states, 'suggests that Sky are more energetically favorable than Asky'. However, this is only valid under certain parameter ranges, as in this work, which is not universal. The author may want to modify the sentence to avoid confusion.

#8. By the end of line 63, I guess the author wants to express "Even after the magnetic field is turned off...", rather than "Once the magnetic field is turned off...".

#9. In the figure caption of Fig. 4, it shows that the temperature for the measurements in Fig. 4a is 295K, and Fig. 4c is 368K. What's the temperature for Fig. 4b? The procedure of applying heat gradients is a bit confusing.

Reviewer #3:

Remarks to the Author:

Heat current-driven transformation of antiskyrmions, non-topological bubbles and skyrmions while under a magnetic field and at zero field were observed by authors. The difference was found by them but the theoretical Interpretation was insufficient or say was lacked in the current state as was admitted by authors themselves as well. The experiment is complex to be honest, for example, the results shown in Figure 4 was taken after heater current applications, a cooling process was also occurred in this experiments I think. Therefore, I suggest a minor revision, and I suggest the paper was modified and resubmitted or was transfer to communication materials.

In the Lorentz images of this manuscript, it is obvious that there is a contrast of black or white dots existing in skyrmions and antiskyrmions, which may be not consistent with conventional DMI skyrmions in FeGe or antiskyrmions in $\text{Mn}_{1.4}\text{PtSn}$ system, Can the authors explain this?

Furthermore, the TIE analysis in Figure 1e,f and schematics of antiskyrmions and skyrmions may illustrate that there is not extra contrast in the center of these two objects as well, please confirm this!

Response to Reviewer's Comments

First, we thank the reviewers for taking their time to review our manuscript, providing constructive comments for us to improve the quality of our work. We have carefully reviewed these comments and have incorporated them into the revised manuscript. Please find our point-by-point response to each review below. The original reviews are shown in *blue italics*. Additionally, we've **highlighted** all modifications within the revised manuscript.

Reviewer Comments:

Reviewer #1:

Yasin et al. performed an experimental investigation of antiskyrmion-hosting ferromagnet (Fe_{0.63}Ni_{0.3}Pd_{0.07})₃P under the influence of thermal gradients. They showed that the application of a thermal gradient drives the transformation of antiskyrmions into non-topological solitons and then skyrmions. The opposite transformations occur starting from a skyrmion lattice. Micromagnetic simulations are also performed to confirm the experimental results. The paper is well-written, well-organized and easy to read. The topic of combination of thermal gradients and solitons is very timely, and the results achieved are interesting. However, I am not convinced that the results are due to the application of the thermal gradients, as also the authors state in the papers. Therefore, I cannot recommend the publication unless the authors unequivocally prove the key role of the thermal gradients, by addressing the following comments:

We thank the reviewer for their comments and are pleased with their positive assessment of our manuscript. We would like to emphasize that the magnonic torque-driven (thermal gradient, ∇T -driven) motion of magnetic defects drives the alignment of the magnetic helices into a single q -domain state as shown from both experiment and in simulation. Additionally, ∇T drives the zero-field, unidirectional topological transformation from skyrmion to antiskyrmion. We also observed the latter phenomena during uniform heating experiments. These results indicate that antiskyrmions are robustly metastable, and that if the local temperature doesn't exceed some critical thresholds described in the main text, thermal gradients are sufficient to drive the transformation, a detail critical to the design of future devices employing thermal gradients. We have since estimated that the temperatures of the device in the field of view observed in the ∇T -driven zero field experiments range from $T = 343$ K to $T = 368$ K at the highest ∇T value (Fig. 3c, formerly Fig. 4c) without a significant decrease in antiskyrmion population. In contrast, the antiskyrmion population begins to decrease at $T = 330$ K during the uniform heating experiment (Fig. 4d, formerly Fig. 5d), suggesting that while both ∇T and uniform heating can trigger the transformation from skyrmions into antiskyrmions, they differ quantitatively. The temperature gradient uniquely contributes to the robust metastability of antiskyrmions at zero field and also drives the motion of both skyrmions and antiskyrmions as we show later in this response. These zero-field, room temperature experiments are the first of their kind, and we therefore believe it is valuable to include both to illustrate the experimental reality which could be the basis for future theoretical work elaborating on these phenomena. We hope that the reviewers agree with our assessment of the data.

1. Paragraph DT-driven helical q-vector switching

The results of Fig. 2 should be confirmed both in the experimental system and simulations. Authors should prove that that transformation does not occur with small increase of a uniform temperature but only under a thermal gradient.

We thank the reviewer for commenting on this important point. We have performed new experiments where we uniformly heat the thin plate pictured in Extended Data Fig. 1b (previously Fig. S2b) from $T_0 = 293$ K to $T = 393$ K before cooling it back to $T_0 = 293$ K at heating/cooling rates of $\frac{dT}{dt} = 5$ K s⁻¹, 10 K s⁻¹, and 50 K s⁻¹, which is the same order of magnitude as the heating and cooling rates of the ∇T experiment shown in Fig. 2 of the main text. As shown in Fig. R1 below (and included in the supplemental material as the new Fig. S2), the initial double- \mathbf{q} vector helical state reemerges upon cooling the sample for all three heating/cooling rates. This implies that the alignment of the double- \mathbf{q} vector state into a single- \mathbf{q} vector state must be driven by a ∇T -related mechanism, but not by thermal fluctuations (uniform heating), such as the one we hypothesize in the current work. We have added this latter discussion to the supplementary material, and added the following text to the third paragraph of the section titled “ **∇T -driven helical \mathbf{q} -vector switching**.”

“Additionally, we performed an experiment in which we uniformly heated the two \mathbf{q} -vector helical state above T_C and then cooled to room temperature, resulting in the return of the two \mathbf{q} -vector helical state as shown in Fig. S2.”

Figure R1 (Fig. S2): Effect of uniform heating on magnetic helical states in (001) $(\text{Fe}_{0.63}\text{Ni}_{0.3}\text{Pd}_{0.07})_3\text{P}$ thin plate. a-i, LTEM micrographs (a, d, g) before, (b, e, h) during, and (c, f, i) after uniform heating from $T_0 = 293$ K to $T_f = 393$ K, and then cooling back to $T_0 = 293$ K at a rate of (a-c) $dT/dt = 5$ K s^{-1} , (d-f) $dT/dt = 10$ K s^{-1} , and (g-i) $dT/dt = 50$ K s^{-1} . The initial double q -vector helical state returns upon cooling below T_C . Scale bars 1 μm .

2. Paragraph DT-driven topological transformation of Asky to Skys at finite field. Authors say: “Indeed, a transition from Asky to NTBs to Sky is observed when the thin plates are heated uniformly consistent with the previously reported $(\text{Fe}_{0.63}\text{Ni}_{0.3}\text{Pd}_{0.07})_3\text{P}$ magnetic state diagram¹².” Then, what is the significance of the thermal gradient? The key role of the thermal gradient should be clearly evident, otherwise the achievements are only incremental results of previous works cited by the authors.

We thank the reviewer for this important comment and the chance to clarify. As we state in the text, we believe the transformation from Asky to NTBs to Sky while under the influence of a temperature gradient is due to thermal fluctuations rather than the temperature gradient. However,

these results confirm that a temperature gradient is sufficient to drive the transformation at a finite field. With the help of micromagnetic simulations, we were able to explain this transformation in terms of thermal fluctuations. However, we believe this study would not be complete without the inclusion of data probing the ∇T -induced phenomena at several fields and with several initial conditions. To address the reviewer's concern, we have moved Fig. 3 to the Extended data, now Extended Data Figure 2.

3. Paragraph DT-driven topological transformation of Asky to Skys at finite field. I find this transformation quite obvious considering the field protocol used to stabilize skyrmions at zero field. Skyrmions remain trapped in the high-field metastable state due to the sudden field removal. The following application of larger temperature, restores the ground state made by helices.

Again, which is the role of the thermal gradient compared to uniform increased temperature?

Indeed, authors commenting the results of Fig. 5 say that is not the consequence of thermal gradient "This suggests that the DT-driven transformations shown in Fig. 4 at zero field may be driven in fact by thermal fluctuations, rather than via magnonic or spin transfer torque"

We thank the reviewer for the observation and would like to take this opportunity to summarize the results carefully. Figures 3 and 4 (formerly Figs. 4 and 5) concern the unidirectional topological transformation from Skys to Askys at *zero* field. The reviewer is correct that the skyrmions remain trapped in a high-field metastable state due to the sudden field removal. Indeed, the result would be quite trivial had the skyrmions annihilated into the helical ground state, but they instead unidirectionally transform into antiskyrmions! We observed this result for thin plate devices under a temperature gradient and under uniform heating, again indicating that the temperature gradient is a sufficient condition for the transformation to occur. Additionally, we have estimated the temperatures of the device within the field of view while applying ∇T and have added these to the Fig. 3 (formerly Fig. 4) caption. The temperatures range from $T = 343$ K to $T = 368$ K at the highest ∇T value (Fig. 3c, formerly Fig. 4c) without a significant decrease in antiskyrmion population. In contrast, the antiskyrmion population begins to decrease at $T = 330$ K during the uniform heating experiment (Fig. 4d, formerly Fig. 5d), suggesting that while both ∇T and uniform heating can trigger the transformation from skyrmions into antiskyrmions, the temperature gradient notably contributes to the robust metastability of antiskyrmions at zero field and also drives the motion of both skyrmions and antiskyrmions. To express this explicitly, we have added the following text to the third paragraph of the section titled " **∇T -driven topological transformation of Skys to Askys at zero field.**" We've included the modified text here for the reviewer's convenience.

"b-c, LTEM micrographs of the zero-field magnetic states taken after heater current applications of (b) $\nabla T = 2.0 \text{ K} \cdot \mu\text{m}^{-1}$ ($314 \text{ K} < T_{FOV} < 323 \text{ K}$, where T_{FOV} is the temperature in the field of view) and (c) $\nabla T = 5.2 \text{ K} \cdot \mu\text{m}^{-1}$ ($343 \text{ K} < T_{FOV} < 368 \text{ K}$) lasting 15 s (except for those values noted in the Methods)."

"In other words, ∇T is a sufficient, but not necessary condition for this topological transformation to occur. Importantly though, the temperature range in the field of view at the highest ∇T value

(Fig. 3c) is $343 \text{ K} < T_{FOV} < 368 \text{ K}$, and at this value, we didn't observe a significant decrease in antiskyrmion population. In contrast, the antiskyrmion population begins to decrease at $T = 330 \text{ K}$ during the uniform heating experiment (Fig. 4d), suggesting that while the two experiments are qualitatively similar, they differ quantitatively, and the temperature gradient contributes to the antiskyrmions' robust metastability at zero field."

4. Also, I did not understand if the transformations in Fig. 5 start from the same initial configuration of Fig. 4. Fig. 5b is not the same as Fig. 5a, while, to prove the authors result, the initial state must be the same and check how it evolves with a uniform increasing temperature.

We thank the reviewer for the nice comment. The reviewer points out that Fig. 5b is not the same as Fig. 5a, which is true because Fig. 5b is taken after heating the sample to $T = 310 \text{ K}$. The LTEM micrograph displayed in Fig. 5a was acquired after reducing the magnetic field to zero while holding the temperature constant at $T = 300 \text{ K}$, and therefore is the initial state. In this case, as seen in the micrograph, the initial state consists of metastable Skys, NTBs, and Askys embedded in a helical background state. Indeed, this is the same metastable state as seen in the LTEM micrograph displayed in Fig. 4a, albeit with slightly different initial spin texture populations. Since these textures are metastable states that randomly emerge/decay during the protocol, it is impossible to reproduce the exact same initial state in subsequent experiments. The fact that the result remains the same while varying the initial spin texture populations suggests that the transformation is repeatable and general. To clarify this point, we've added the following text to the third paragraph of the section titled ' ∇T -driven topological transformation of Skys to Askys at zero field.'

"The initial metastable state shown in Fig. 5a was prepared in the same way as in Fig. 4a, and consists of the same mixture of metastable Skys, NTBs, and Askys, although with slightly different initial populations."

5. Why no skyrmion/antiskyrmion motion is observed? Such as in Phys. Rev. B 106, 024415, July 2022.

We thank the reviewer for the great observation. Indeed, we have plans to study the temperature gradient-driven motion of topological spin textures, and have preliminary data shown in the figure below. However, we believe that it is beyond the scope of the current study, which focuses on the transformations between different magnetic states, rather than the motion of skyrmions or antiskyrmions.

Figure R2: Temperature gradient-driven motion of skyrmions and antiskyrmions in a $(\text{Fe}_{0.63}\text{Ni}_{0.3}\text{Pd}_{0.07})_3\text{P}$ thin plate. **a-b**, LTEM micrographs taken at time (a) $t = 0$ s and (b) $t = 17.75$ s with four elliptical skyrmions labelled and their ∇T -driven motion tracked by solid line plots. **c-d**, LTEM micrographs taken at time (a) $t = 0$ s and (b) $t = 18.625$ s with six elliptical skyrmions labelled and their ∇T -driven motion tracked by solid line plots. The micrographs were obtained via live recording with an 8 fps framerate with (a-b) $\nabla T = 0.4 \text{ K} \mu\text{m}^{-1}$ and (c-d) $\nabla T = 1.2 \text{ K} \mu\text{m}^{-1}$. **e**, Skyrmion velocity v and Hall angle θ_{Hall} plotted versus ∇T using open and closed circular markers and solid black lines and dashed green lines, respectively. Additionally, the antiskyrmion velocity and Hall angle are plotted versus ∇T using open and closed circular markers and solid black lines and dashed green lines, respectively. Scale bars 1 μm .

6. Also, apart from the phase transitions - the same group has shown similar phase transitions in a previous work Ref. 12 - , which is the importance of these results for applications? Motivation,

significant advancements in the field as well as a perspective are only slightly mentioned or missing.

We thank the reviewer for this important remark. Indeed, Ref. 12 maps out the magnetic phase diagram in Fig. 2g of that work. We would like to emphasize that the current study reports two phenomena that are unexplainable by that earlier work:

First, the ∇T -driven switching of the helical q-vector states, which has never been reported before and demonstrates the magnonic torque-driven motion of magnetic defects as described in the main text.

Second, the ∇T -driven unidirectional transformation from Skys to Askys at *zero field*. The results of Ref. 12 suggest that instead of Askys, metastable Skys at zero field would transform into magnetic helices, the ground state. The inexplicability of these results highlights the significance and need for the current study. To emphasize this point and discuss an explanation of these results, we've added the following text to the third paragraph of the section titled ' ∇T -driven topological transformation of Skys to Askys at zero field.'

"We performed micromagnetic simulations at zero field for Skys, Askys and helical states while varying temperature and found that the Skys and Askys would relax into the helical state within our model, which agrees with the previously published literature¹². This contradicts the experimental findings shown here and suggests that more extensive investigations and perhaps new theory are necessary to explain this delicate process.

Let us outline an argument explaining this result qualitatively. To change the topology of the magnetic texture, one needs to introduce a topological defect called Bloch point or anti-Bloch point (or magnetic hedgehog), a singularity in the magnetization around which the magnetization points in all possible directions. At this topological defect the magnetization changes rapidly in space which costs a lot of exchange energy. Accordingly, we can expect that the energy of a Bloch point inside the closed domain wall that constitutes an (anti)skyrmion/bubble is lower than its energy in the polarized background between textures as the magnetization inside the domain wall is already twisted. Thus, a Bloch point is more likely to emerge in the textures' domain walls, rendering topological transitions between isolated defects such as in the transition from Sky to NTB to Asky observed here, rather than the transition to the helical state which requires the fusion of neighbouring defects via a Bloch point in the polarized background space between the texture and either its neighbour or the helical background."

Other comments:

- The motivation should be expanded for a broad audience. Why is it important to use thermal manipulation of solitons over other methodologies, such as electrical?

We thank the reviewer for the opportunity to improve our work. We have expanded the second paragraph of the introduction to specifically motivate our study of the thermal manipulation of magnetic spin textures. Additionally, we have added reference 18: Rattner, A. S. & Garimella, S. Energy harvesting, reuse and upgrade to reduce primary energy usage in the USA. *Energy* **36**, 6172–6183 (2011).

“The prototypical stimulus used to manipulate and control **electron-spins** is electric current, and several studies regarding spin texture dynamics have thus been undertaken including current-driven domain wall motion¹⁵, helical domain modulation wave vector alignment¹⁶, Sky lattice motion⁶, and single Sky motion¹⁷. **However, as energy demands increase across the world, interest in technology capable of harvesting the waste heat generated from energy conversion processes such as those in automobiles and thermal power plants has spiked, as an estimated two thirds of the required input energy is converted to heat¹⁸. As such, the identification of heat-driven processes and materials that enable ‘green’ spintronics is critically important. One such process is the thermal current-driven domain wall^{19,20} and Sky motion²¹, and recent experiments have confirmed such Sky motion in both insulating²² and metallic^{23,24} magnets. A recent study has also found that unique combinations of externally applied magnetic field and heat may drive the controlled, reversible topological transformations between Skys and Askys in Asky-hosting magnets^{12,13}. In this study, we designed a focused ion beam (FIB) - fabricated device (see Fig. S1 in the supplemental information) to apply a range of temperature gradients ∇T across the room temperature ($T_C \approx 362\text{K}$, see the magnetic state diagram in Fig. S2a), noncentrosymmetric tetragonal crystal ($\text{Fe}_{0.63}\text{Ni}_{0.3}\text{Pd}_{0.07}$)₃P and measure the in-situ response of the native magnetic helices, (anti)skyrmions and NTBs in real space using Lorentz transmission electron microscopy (LTEM). In the following, we summarize the ∇T -driven transformations observed in the thin plate magnet and then discuss the potential mechanisms for each evidenced by the experimental data.”**

- Figure 2b is unclear. Where is the Pt wire? What is the white rectangle? Which are the dimensions of the sample in the inset and in the main panel?

We thank the reviewer for noticing this important point. We have modified the caption to include “**The inset outlined by dashed red lines in (b) is a scanning electron micrograph taken with a 52° sample tilt and indicates the field of view ($3.2 \times 3.2 \mu\text{m}^2$, outlined by dashed black lines) of the experiment (a-c), in which $T > T_C$.**” As noted in the caption already, the Pt wire is on the right-hand-side (outside) of the images, and the white rectangle is a scale bar, which is 10 μm . We have re-cropped the inset image in both Fig. 2 and the new Fig. 3 (previously Fig. 4) to include more of the Pt wire so it is more obvious to the reader. The modified Figures are included below.

Figure R3 (Fig. 2): Heat current-driven ordering of a double q -vector helical state to a single q -vector alignment at zero field. **a-c**, LTEM micrographs of the (a) initial double q -vector helical state at 295 K, (b) paramagnetic state during the application of $\nabla T \approx 4.2 \text{ K} \cdot \mu\text{m}^{-1}$ which causes an increase in the base temperature to between 369 K and 382 K in the field of view and (c) resulting single q -vector helical state upon termination of the heater current and cooling of the field of view to a uniform 295 K in $\approx 1 \text{ s}$. **The inset outlined by dashed red lines in (b) is a scanning electron micrograph taken with a 52° sample tilt and indicates the field of view ($3.2 \times 3.2 \mu\text{m}^2$, outlined by dashed black lines) of the experiment (a-c), in which $T > T_c$.** **d-g**, Micromagnetic simulations of ∇T -driven ordering of the magnetic domains from the (d) initial double q -vector helical state to a (e) paramagnetic state during the application of scaled temperature gradient $\nabla T_1^* = 20 \text{ K} \cdot \mu\text{m}^{-1}$, (f) partially aligned double q -vector helical state during the application of $\nabla T_2^* = 4.9 \text{ K} \cdot \mu\text{m}^{-1}$ and (g) resulting single q -vector helical state after applying ∇T_2^* for a sufficiently long time. A magnetic defect in the helical state shown in (f) is indicated by an arrow. The heater is located on the right-hand-side of each image, while the cold bath is on the left-hand-side. Scale bars are $1 \mu\text{m}$ in (a-c) and $10 \mu\text{m}$ in the inset in (b).

Figure R4 (Fig. 3). Unidirectional transformation of topological spin textures at zero field via heat current. **a**, LTEM micrograph of the initial magnetic state composed of metastable spin textures stabilized by applying B_z to form a Sky lattice state and then reducing the field to zero at 295 K (schematic on LHS of **a**), after which some skyrmions remained stable. **b-c**, LTEM micrographs of the zero-field magnetic states taken after heater current applications of **(b)** $\nabla T = 2.0 \text{ K} \cdot \mu\text{m}^{-1}$ ($314 \text{ K} < T_{FOV} < 323 \text{ K}$, where T_{FOV} is the temperature in the field of view) and **(c)** $\nabla T = 5.2 \text{ K} \cdot \mu\text{m}^{-1}$ ($343 \text{ K} < T_{FOV} < 368 \text{ K}$) lasting 15 s (except for those values noted in the Methods) and raising the temperature in the field of view up to 368 K. Selected spin textures are marked as they transform from Sky (blue arrows) to NTBs (green arrows) to Askys (red arrows). **d**, Spin texture population (N_{ST} is the combined number of antiskyrmions Asky, nontopological bubbles NTB, and skyrmions Sky) versus ∇T of the full $7.3 \times 5.5 \mu\text{m}^2$ field of view in which the $4.6 \times 4.6 \mu\text{m}^2$ area shown in **a-c** is contained. Five types of spin textures are identified, including Askys (red square markers), horizontal Skys ('H Sky' with open blue triangle markers), vertical skyrmions ('V Sky' with blue, upside-down triangle markers), horizontal non-topological bubbles ('H NTB' with open green equal-length-diagonal rhombus markers), and vertical non-topological bubbles ('V NTB' with green rhombus markers). The total number of spin textures is plotted by a solid black line with open circular markers. The inset panel shows the field of view of the experiment within the $(\text{Fe}_{0.63}\text{Ni}_{0.3}\text{Pd}_{0.07})_3\text{P}$ device, and the right-hand-side legend shows schematics of the in-plane magnetization for each spin texture measured in **a-d**. Scale bars in **a-c** are $1 \mu\text{m}$ and $10 \mu\text{m}$ in inset panel of **d**.

- Which is the temperature dependence of M_s in the simulations? It should be clearly added to Methods.

We thank the reviewer for bringing up this point about our simulations which use different techniques for Figs. 2 and Extended Data Fig. 3 (formerly Fig. 3). This seems to be confusing for the readers. We have modified the Methods section accordingly and put more emphasis on the fact that we are using different methods for these two cases. The reason is the following:

The process in Fig. 2 is a dynamical process which requires simulations that take into account the full dynamics, using a stochastic (Langevin) Landau-Lifshitz-Gilbert simulation. These types of simulations require a high numerical lattice resolution in order to mimic atomistic spin dynamics with temperature independent atomistic interactions and material parameters. The effect of temperature is modelled by a fluctuating magnetic field which locally satisfies the fluctuation-dissipation theorem. The micromagnetic parameters such as M_s , however, are coarse-grained versions of the atomistic parameters and therefore become temperature dependent because of the fluctuating magnetic field. We do not make this clear distinction between effective parameters in the Methods section as this is usually also neglected in other works which follow this approach. In short: The effective M_s is indeed temperature dependent, but this is not important for the method as it is automatically deduced from a temperature independent M_s combined with a Langevin force (fluctuating field) in the simulations. Note that we did not use parameters for $T = 0$ K but $T = 300$ K to account for the thermal fluctuations below the resolution of our numerical simulation grid.

The process in Extended Data Fig. 3 is very different as there we are tracing static properties of the textures, i.e., an approximation for the free energy. This is achieved by considering a micromagnetic model with micromagnetic parameters that are temperature dependent, i.e., the action of the above-mentioned fluctuating field is integrated out and absorbed into the parameters which thereby become dependent on M_s . Therefore, we can plot the lowest energy textures in Extended Data Fig. 3 as a function of M_s . The exact dependence of M_s on the temperature is, however, beyond the scope of this model. The only relevant relation between M_s and T is that for increasing T the M_s decreases.

- Since many combinations of field and temperature are considered in the main text, I suggest to move the phase diagram of Fig. S2a to the main text and make it clearer. Authors can mark the phase diagram points where the images are taken. Skyrmion lattice should be also highlighted. FM and PM regions should have different colors.

We thank the reviewer for this important comment. We have added Fig. S2 to the main text as the new Extended Data Figure 1. We have shaded the paramagnetic region purple to make it visually distinct from the ferromagnetic region. As noted in the caption, the circular markers are the data points where the images are taken. We've included the figure below.

Figure R5 (Extended Data Figure 1): Magnetic state diagram in a $\text{Fe}_{1.9}\text{Ni}_{0.9}\text{Pd}_{0.2}\text{P}$ thin plate.

a, Density of spin textures as a function of temperature T [K] and applied magnetic field B [mT]. The spin textures in the measurement include antiskyrmions (Asky), nontopological bubbles (NTB), and skyrmions (Sky). Spin texture density is represented by the colormap shown on the right-hand-side, with dark blue representing $\rho_N = 0$ (lowest energy helical magnetic state) and dark red representing the maximum ρ_N observed in the thin plate, corresponding to the skyrmion lattice state (SkyL). The purple area shows the paramagnetic state (PM) at $T > T_c \approx 362$ K, while the white area shows the ferromagnetic state (FM) in which the spins are polarized along the same direction as the applied magnetic field when applied above the saturation field illustrated by the boundary drawn by the solid black line. The circular markers represent the temperature and field values at which real space LTEM images were acquired to measure the spin texture density and magnetic states present. **b**, SEM micrograph showing a top-down view of the thin plate which is $\approx 12 \mu\text{m} \times 21 \mu\text{m}$ in area and has a thickness of ≈ 150 nm, which is approximately the same size as the thin plates in the temperature gradient devices. Scale bar $10 \mu\text{m}$.

- Fig. 4a looks more like a mixed phase of skyrmions and helices rather than a skyrmion lattice.

We thank the reviewer for letting us explain this important point further. Although a skyrmion lattice state was thermodynamically stable when we created it with an external magnetic field, the state changed once we turned the field off, and only some skyrmions survived in a cluster state, as we mention in the figure caption. We have also added “The resulting metastable Skys (with some NTBs and Askys) are shown in Fig. 4a...” to the first paragraph of the section titled ‘ ∇T -driven topological transformation of Skys to Askys at zero field.’ in the main text to further clarify this point.

- In general, the experimental figures lack resolution quality, especially when compared with the images in Ref. 12 from the same group. Is it possible to increase images definition?

We thank the reviewer for allowing us to explain this point. In Ref. 12, all the images were static images, and so the exposure time could be increased until sufficient contrast was attained. In turn, in the present experiments, we applied external stimuli and observed the spin texture’s response using live recording. The resulting images are therefore frames with an exposure time of 0.125 s. Unfortunately, this is unavoidable in this case. However, we believe that this unfortunate issue is at present of purely aesthetic value and that our main findings are still clearly visible in the data.

Reviewer #2:

In the manuscript “Heat current-driven topological spin texture transformations and helical q-vector switching”, the authors study how heat gradient could trigger the switching between different topological magnetic nanostructures (which are antiskyrmions, non-topological bubbles, and skyrmions) in the material system with S_4 point group crystal symmetry. The material system with S_4 (as well as D_{2d}) symmetry is a novel new material system with anisotropy DMI interaction, thus enabling magnetic nanostructure with different topologies to be stabilized, which is a novel platform for the non-volatile memory device. Then the question to the community is how to control switching between these metastable spin structures. In their previous work (ref. 12, 13), the authors show that with the in-plane field induced by tilting the sample, such switching could be achieved. In this work, the authors further show that using heat gradient could also achieve such switching. This should be the first experiment attempt that uses heat gradient to control spin topology switching in the S_4/D_{2d} -based material system, which is clear progress in skyrmion manipulation.

In the manuscript, the following experiment results support the author’s conclusion:

- A. After applying the heat gradient, the original 2-q helix state becomes a single-q helix state, and the single helix state direction corresponds to the heat gradient direction, as shown in Fig. 2a-c.*
- B. With the heat gradient, magnetic defects move along the heat gradient direction, from cold to hot, as shown in Extended Data Fig. 1.*
- C. With the field applied, under the suitable heat gradient, antiskyrmion switches to the non-topological bubble and finally into skyrmion, as shown in Fig. 3a-c.*
- D. Without field applied, under the suitable heat gradient, skyrmion switches to the non-topological bubble and finally into antiskyrmion, as shown in Fig. 4a-c.*
- E. The difference between Fig. 4d and Fig. 5d shows that the effect is not induced purely by uniform heat that heat gradient also contributes.*

The underlying physics, especially how the heat gradient generated torque achieves the switching, could be very interesting and complicated. This work, on the one hand, further confirms the metastability of the spin texture D_{2d}/S_4 system. On the other hand, demonstrates the possibility of using heat current to control skyrmion topology, which might encourage more heat gradient-driven skyrmion dynamic studies. So, I recommend this manuscript be published in Nature Communications with some possible modifications, as commented below:

We thank the reviewer for their comments and are pleased with their highly positive assessment of our manuscript. We agree that this is the first experiment using a temperature gradient to control spin topology switching and helical q-vector switching. Additionally, it’s the first experiment using uniform heating to control the real space spin topology switching at zero field. We agree that this work will encourage more heat gradient-driven skyrmion dynamic studies.

#1. There are two temperatures of relevance: temperature on the cold side of the device and temperature difference across the device. As shown by Extended Data Fig. 2, the authors can nicely calibrate the temperature. Could the authors state the two temperatures of each measurement clearly? For example, in line 60, the cold side is 295K; In line 82, the temperature range is 369K-382K, so I guess the cold side is 369K; In line 88-90, the simulated temperature gradient is given, but the temperature at the cold edge seems not given.

We thank the reviewer for pointing out this very important issue. Such information is of course vital for understanding and reproducing our results. Please allow us to clarify this point. As the reviewer points out, the cold side of the field of view shown in Fig. 2b is 369 K. On the other hand, the cold side of the sample contacted to the Si chip substrate (outside of the field of view) is always held at 295 K. We have clarified this point now in multiple places in the text:

“Upon application of $\nabla T \approx 4.2 \text{ K} \cdot \mu\text{m}^{-1}$ (see the Methods section for a description of how we quantify ∇T), the magnetic contrast disappeared as shown in Fig. 2b. This is because the field of view in Fig 2a-c is a region of the sample located near the Pt heater wire (inset Fig. 2b), where the temperature for $\nabla T \approx 4.2 \text{ K} \cdot \mu\text{m}^{-1}$ ranges from 369 – 382 K. As this temperature range is above T_C , the magnetic state in Fig. 2b is paramagnetic.”

Additionally, we’ve added details to the methods regarding the temperatures within the simulations in Fig. 2. The cold side of the simulated field of view is held at $T_{cold} = 0.95 T_C$. We’ve included the new text here as well for the reviewer’s convenience:

“Micromagnetic simulations

We performed micromagnetic simulations based on the (stochastic) Landau-Lifshitz-Gilbert (sLLG) equation using the MuMax3 software in two distinct fashions for the data shown in Fig. 2 and Fig. 3, respectively.²⁹ For Fig. 2, the simulated magnet size was $L_x \times L_y \times L_z = 4096 \times 2048 \times 128 \text{ nm}^3$ discretized into $256 \times 128 \times 8$ cuboids. The 256 slices were held at different temperatures to achieve $\nabla T \neq 0$ across the magnet, and we set open boundary conditions. The temperature in this case was modelled via a fluctuating magnetic field as implemented in Mumax3. We can express the Curie temperature as $T_C \approx 2 A a_L / k_B$,²⁶ where A is the exchange stiffness, a_L is the lattice constant and k_B is Boltzmann’s constant. We set the temperature on the cold side of the simulation in Fig. 2 to be $T_{cold} = 0.95 T_C$, with $\nabla T_1 = 5 T_C / L_x$ (Fig. 2e) and $\nabla T_2 = 0.05 T_C / L_x$ (Fig. 2e). The material parameters were chosen according to Ref. 12 at $T = 300 \text{ K}$: saturated magnetization $M_S = 4.17 \times 10^5 \text{ A m}^{-1}$; exchange stiffness $A = 8.1 \times 10^{-12} \text{ J m}^{-1}$; anisotropic DMI constant $D_{ani} = 2.0 \times 10^{-4} \text{ J m}^{-2}$ and uniaxial anisotropy constant $K_u = 3.1 \times 10^4 \text{ J m}^{-3}$ along the z axis. The Gilbert damping constant was set to $\alpha = 0.03$. To achieve anisotropic DMI, we added custom energy density and effective field terms using the MuMax3 built-in functions. The field term is of the form $\mathbf{B}_{D_{ani}} = 2D_{ani} \left(-\frac{\partial n_z}{\partial y} \hat{x} - \frac{\partial n_z}{\partial x} \hat{y} + \left(\frac{\partial n_y}{\partial x} + \frac{\partial n_x}{\partial y} \right) \hat{z} \right)$, where n_i is the i^{th} component of the magnetization, and the energy density expression is defined as $E_{D_{ani}} = -\frac{1}{2} \mathbf{n} \cdot \mathbf{B}_{D_{ani}}$.^{13,14,29”}

#2. Due to the similarity in sample composition, I assume the sample used in this paper is the same as the one used in the author’s previous work (Ref. 12). In Fig. 1h of Ref. 12, the M_S have dramatical dependence of temperature when the temperature is above $\sim 350\text{K}$, which is the temperature range studied in this work. So, is the heat gradient-induced switching of skyrmion topology only occur just below T_C ?

We thank the reviewer for catching this point. The sample composition is indeed the same as in Ref. 12. However, the heat gradient-induced switching of skyrmion topology studied here occurs mainly below 350 K. For example, in Extended Data Fig. 3 (formerly Fig. 3) with a finite applied

external magnetic field, $\nabla T = 0.33 \text{ K } \mu\text{m}^{-1}$, which corresponds to a maximum temperature of $T_{max} \approx T_{cold} + \nabla T \times L \approx 302 \text{ K}$, where T_{cold} is room temperature, and $L \approx 21 \mu\text{m}$ is the length of the sample from the cold side to the heater measured using Fig. S1. Our observations at zero field shown in Fig. 3 (previous Fig. 4) indicate that the topological transformations from skyrmion to antiskyrmion begin at $\nabla T \approx 2 \text{ K } \mu\text{m}^{-1}$. The hot side of the field of view is $13.9 \mu\text{m}$ from the cold bath (held at T_{cold}), which corresponds to a maximum temperature in the observed region of $T_{max} \approx 337 \text{ K}$.

We've added the estimated temperature values to the captions of the figures in the main text.

#3. In line 91, the simulated temperature gradient is chosen to be 4.9K/um, where helical order returns. In line 80, the experiment temperature gradient is 4.2K/um (which is lower than 4.9K/um), where the magnetic contrast disappears due to the sample temperature going above T_c . The absolute temperature used in the micromagnetic simulation usually does not have a one-to-one correspondence to the real experiment value. For example, its effect also depends on the simulated cell size chosen. The authors may want to modify this part to avoid confusion.

We thank the reviewer for this nice observation. We agree that this may be confusing, so we've added text to the methods section to indicate how we calculated T_C in simulation and how we rescaled the simulated temperature gradients to be understandable to the readers. We've included the new text here as well for the reviewer's convenience:

“Micromagnetic simulations

We performed micromagnetic simulations based on the (stochastic) Landau-Lifshitz-Gilbert (sLLG) equation using the MuMax3 software in two distinct fashions for the data shown in Fig. 2 and Fig. 3, respectively.²⁹ For Fig. 2, the simulated magnet size was $L_x \times L_y \times L_z = 4096 \times 2048 \times 128 \text{ nm}^3$ discretized into $256 \times 128 \times 8$ cuboids. The 256 slices were held at different temperatures to achieve $\nabla T^* \neq 0$ across the magnet, and we set open boundary conditions. The temperature in this case was modelled via a fluctuating magnetic field as implemented in Mumax3. We can express the Curie temperature as $T_C^* \approx 2 A a_L / k_B$,²⁶ where A is the exchange stiffness, a_L is the lattice constant and k_B is Boltzmann's constant. We set the temperature on the cold side of the simulation in Fig. 2 to be $T_{cold}^* = 0.95 T_C^*$, with $\nabla T_1^* = 5 T_C^* / L_x$ (Fig. 2e) and $\nabla T_2^* = 0.05 T_C^* / L_x$ (Fig. 2e). The material parameters were chosen according to Ref. 12 at $T = 300 \text{ K}$: saturated magnetization $M_S = 4.17 \times 10^5 \text{ A m}^{-1}$; exchange stiffness $A = 8.1 \times 10^{-12} \text{ J m}^{-1}$; anisotropic DMI constant $D_{ani} = 2.0 \times 10^{-4} \text{ J m}^{-2}$ and uniaxial anisotropy constant $K_u = 3.1 \times 10^4 \text{ J m}^{-3}$ along the z axis. The Gilbert damping constant was set to $\alpha = 0.03$. To achieve anisotropic DMI, we added custom energy density and effective field terms using the MuMax3 built-in functions. The field term is of the form $\mathbf{B}_{D_{ani}} = 2D_{ani} \left(-\frac{\partial n_z}{\partial y} \hat{x} - \frac{\partial n_z}{\partial x} \hat{y} + \left(\frac{\partial n_y}{\partial x} + \frac{\partial n_x}{\partial y} \right) \hat{z} \right)$, where n_i is the i^{th} component of the magnetization, and the energy density expression is defined as $E_{D_{ani}} = -\frac{1}{2} \mathbf{n} \cdot \mathbf{B}_{D_{ani}}$.^{13,14,29}

#4. There is a difference in DMI between the S4 and the D2d system. The S4 point group has generators of $(-x,-y,z)$ and $(y,-x,-z)$. The D2d point group has one more generator of $(-x,y,-z)$. For the D2d point group, the DMI Hamiltonian is $H_{D2d} = D(My \partial Mz / \partial x - Mz \partial My / \partial x) - D(Mz \partial$

$M_x/\partial y - M_x \partial M_z/\partial y$); For the S_4 point group, the symmetry is lower than that in D_{2d} and the DMI Hamiltonian is: $H_{S_4} = D_1(M_y \partial M_z/\partial x - M_z \partial M_y/\partial x) - D_1(M_z \partial M_x/\partial y - M_x \partial M_z/\partial y) + D_2(M_y \partial M_z/\partial y - M_z \partial M_y/\partial y) + D_2(M_z \partial M_x/\partial x - M_x \partial M_z/\partial x)$. By changing the coordinate system that rotates along the Z-axis for a certain angle regarding D_1 and D_2 , H_{S_4} could transform into the form of $H_{D_{2d}}$. This means that due to the lower symmetry in the S_4 system, the DMI preferred Bloch direction is not exactly along $[100]$ but with a certain angle related to D_1 and D_2 . This angle can be clearly observed in Fig. 2a. The authors should modify the corresponding parts.

We thank the reviewer for bringing up this important point. Indeed, at this level of approximation in powers of gradients and magnetization, S_4 and D_{2d} are identical up to a rotation angle around the z-axis that is not determined by symmetries, i.e., the axes for the Bloch type screws in S_4 are not locked by symmetry, while they are glued to the 100-axes in D_{2d} -symmetric materials. In fact, for this material composition, the Bloch type axes are approximately the 110-directions as we previously pointed out in Ref. 12. We believe that this extra degree of freedom in S_4 materials over D_{2d} materials is another point that makes this material class very interesting and should be further recognized. For this reason, the original version of the manuscript already had coordinate systems in Fig. 1 and Fig. 2 which explicitly indicate the 110-directions and not the 100-directions. As was indicated in Fig. 2b, also the temperature gradient was along one of the 110-directions.

We have edited the main text to stress this point even more and have included the relevant section here for the reviewer's convenience.

“Let us first analyse the zero-field, ∇T -driven transformation of the initial two \mathbf{q} -vector helical state, shown in the defocussed LTEM micrograph in Fig. 2a, with \mathbf{q}_1 and \mathbf{q}_2 almost parallel to the $[\bar{1}10]$ and $[110]$ crystal axes, respectively, consistent with previous observations¹².”

For the simulations we have rotated the coordinate system such that the Bloch type axes are along the x and y axis, respectively, and the temperature gradient is along the x axis. The small mismatch between the temperature gradient and Bloch type axes in the experimental setup, opposed to the perfectly aligned axes in the theoretical setup, is only of minor importance as the angle is very small. To address explicitly, we have added the following line to the Methods section titled “Micromagnetic simulations.”

“In both simulations, the x and y axes correspond to the experimental $[110]$ and $[1\bar{1}0]$ crystal axes, respectively.”

#5. In Fig. 2b, the temperature is 369K-382K which is larger than $T_c=362K$. Is heating the device above T_c a must for switching from the 2-q state to the 1-q state? Since Extended Data Fig. 1 shows that defect could move along the heat gradient, the switching seems could be achieved below T_c already.

We thank the reviewer for allowing us to clarify this important point. While our observations suggest that magnetic defects move along a heat gradient, there is a noticeable lack of magnetic defects present in the ground magnetic helical state shown in Fig. 2a. By raising the temperature

of the field of view above T_C using a finite ∇T and then lowering again, magnetic defects may readily form and then proceed to comb the single \mathbf{q} -vector state. To emphasize this, we've added to the second paragraph in the subsection titled “ **∇T -driven helical \mathbf{q} -vector switching**” and have included our edits here for the reviewers' convenience.

“When the helical ordering returned in Fig. 2f, we observed the formation of several magnetic defects including the dislocation defect marked by an arrow, which began to propagate across the sample from cold to hot due to the magnonic torque induced by ∇T_2^{*27} .”

#6. *The bending contours are different between Fig. 2a and Fig. 2c and between Extended Data Fig. 1 a-d. Why is that? Maybe comes from temperature difference?*

We thank the reviewer for mentioning this point. As the sample heats up due to the applied ∇T , the thin plate will likely experience strain as the lattice slightly expands. This would result in slightly different strain contours manifesting as the dark intensity regions pointed out by the reviewer, as seen in Fig. 2c in particular. These contours shift depending on the magnitude of ∇T as shown in Extended Data Fig. 2a-d (previously Extended Data Fig. 1). After application of a temperature gradient, it takes time for these contours to return to their normal state. The micrograph displayed in Fig. 2c was acquired only 5 – 6 s after termination of the heater current. For comparison, we've included Fig. R6 below, which shows two LTEM micrographs before and after the application of a slightly lower temperature gradient, $\nabla T \approx 3.8 \text{ K } \mu\text{m}^{-1}$. Fig. R6b shows the final state, which acts as the initial state in Fig. 2a. Clearly, the contour contrast has disappeared in the minutes in between the two experiments.

Figure R6. Heat current-driven ordering of a single \mathbf{q} -vector helical state to a double \mathbf{q} -vector alignment at zero field. **a**, LTEM micrograph of the initial magnetic state, a single \mathbf{q} -vector helical state. **b**, LTEM micrograph of the magnetic state after application of $\nabla T \approx 3.8 \text{ K } \mu\text{m}^{-1}$. Scale bars $1 \mu\text{m}$.

#7. *In line 103, the authors confirm the cold-to-hot heat gradient-driven motion of magnetic defects and show it in Extended Data Fig. 1. What's the velocity of this motion? And is the velocity*

dependent on the heat gradient amplitude? Even more interesting is that will antiskyrmion/non-topological bubble/skyrmion also have similar heat gradient-driven motion? And since these topological magnetic nanostructures are anisotropic, will heat gradient be applied along different crystalline directions to trigger different dynamics? This question may go beyond the scope of this paper since heat gradient measurement in LTEM itself is already not easy.

We thank the reviewer for this very interesting question. In this case, it is quite difficult to calculate the velocity of the magnetic defect motion as a function of temperature gradient magnitude because the magnetic defect moved as the temperature gradient was increasing, which is why we only label the position of the defect in Extended Data Fig. 2 (previously Extended Data Fig. 1). As far as ∇T -driven motion of antiskyrmions and skyrmions, we refer to our preliminary data shown in Fig. R2. Indeed, they exhibit motion, although the motion is from hot to cold, consistent with Ref. 23. We have not yet performed experiments in which we apply ∇T along different crystal axes, although this is a lovely suggestion. We agree with the reviewer that such a study is beyond the scope of this paper which focuses on ∇T -driven transformations between spin textures.

#8. The similarity and difference between Fig. 4d and Fig. 5d show that both uniform heat and heat gradients contribute to the final switching behavior. It would be nice if the authors could discuss the underlying physics mechanism.

We thank the reviewer for bringing up this important point. Indeed, there are theoretical methods such as the geodesic nudged elastic band (GNEB) method which can be used to study minimal energy barriers and, thus, the most likely decay paths. However, unfortunately, methods of this kind require multiple copies of the magnetic texture at once. Thus, they are multiple times more memory intensive than simple simulations and memory is a very limited resource on GPUs. Moreover, this method is not yet implemented in MuMax3 which would be a tremendously complex programming task on its own, far beyond the scope of this project.

Instead, we can try to qualitatively argue the mechanism for the topological transformation from skyrmions to antiskyrmions at zero field based on the experimental observations. To change the topology of the magnetic texture, one needs to introduce a topological defect called Bloch point or anti-Bloch point (or magnetic hedgehog), a singularity in the magnetization around which the magnetization points in all possible directions. At this topological defect the magnetization changes rapidly in space which costs a lot of exchange energy. Accordingly, we can expect that the energy of a Bloch point inside the closed domain wall that constitutes an (anti)skyrmion/bubble is lower than its energy in the polarized background between textures as the magnetization inside the domain wall is already twisted. Thus, a Bloch point is more likely to emerge in the textures' domain walls, rendering topological transitions between isolated defects such as in the transition from Sky to NTB to Asky observed here, rather than the transition to the helical state which requires the fusion of neighboring defects via a Bloch point in the polarized background space between the texture and either its neighbor or the helical background. We have included the following text in the fourth paragraph of the section of the main text titled “ **∇T -driven topological transformation of Skys to Askys at zero field.**”

“Let us outline an argument explaining this result qualitatively. To change the topology of the magnetic texture, one needs to introduce a topological defect called Bloch point or anti-Bloch point

(or magnetic hedgehog), a singularity in the magnetization around which the magnetization points in all possible directions. At this topological defect the magnetization changes rapidly in space which costs a lot of exchange energy. Accordingly, we can expect that the energy of a Bloch point inside the closed domain wall that constitutes an (anti)skyrmion/bubble is lower than its energy in the polarized background between textures as the magnetization inside the domain wall is already twisted. Thus, a Bloch point is more likely to emerge in the textures' domain walls, rendering topological transitions between isolated defects such as in the transition from Sky to NTB to Asky observed here, rather than the transition to the helical state which requires the fusion of neighbouring defects via a Bloch point in the polarized background space between the texture and either its neighbour or the helical background."

#9. Under zero field, the helix should be the ground state. In Fig. 4 and Fig. 5, when analyzing the skyrmion/non-topological bubble/antiskyrmion number, are the long-length magnetic topological structure counts as 'skyrmion' etc. or 'short helix'?

We thank the reviewer for allowing us to clarify this point. We classified the spin textures as Sky/Asky/NTB only if the domain walls were contained within the field of view. The spin texture was classified as helical if it extended beyond the field of view of observation. Therefore, it was possible to have some 'long-length' Skys, Askys and NTBs. To indicate this explicitly in the main text, we've added a small section in the methods and have included it here for the referee's convenience.

"Classification of spin texture populations

In the zero-field LTEM micrographs analyzed for Fig. 3 and Fig. 4, we classified the spin textures as either Sky, Asky, NTB or as part of the helical background. The spin textures were classified as helices if they extended beyond the field of view of observation. Therefore, it was possible to have some 'long-length' Skys, Askys and NTBs."

#10. Is the heat gradient effect in this S4 material system similar to the other system reported before, such as Bloch skyrmion in B20?

We thank the reviewer for this important question. To our knowledge, there are no reports of temperature gradient-driven transformations of magnetic helices or topological skyrmions, even in the ubiquitous B20 compounds. Ref. 16 showed a similar combing of the magnetic helix in FeGe, although this transformation was driven using an electric current. Perhaps more interesting are the studies involving temperature gradient driven skyrmion motion, such as that seen in Ref. 24. In that work, the authors found skyrmion motion with a direction that depended on the temperature gradient magnitude. This was due to the temperature gradient driving both the motion of itinerant electrons as well as magnons. Because $(\text{Fe}_{0.63}\text{Ni}_{0.3}\text{Pd}_{0.07})_3\text{P}$ is also a metal, perhaps it will exhibit similar physics in future dynamics studies.

Below are a few minor points:

#1. Ref. 25 is cited in line 90, which is the book that seems not so related to the temperature discussed there.

We thank the reviewer for allowing us to clarify. The temperature scaling we refer to in the second paragraph in the subsection titled “**VT-driven helical q-vector switching**” is from an equation in Ref. 26 (previously Ref. 25). This equation is now explicitly stated in the Methods section, in the subsection titled “Micromagnetic simulations.”

#2. In Fig. 1d, the two helix directions have an angle away from 90 degrees (different than that in Fig. 2a). Is this due to local pinning?

We thank the reviewer for catching this detail. Indeed, this slight difference is likely due to local pinning and or magnetic defects that exist outside of the field of view.

#3. In supplementary video 2, at above 359K, the helix changes its direction. Is it due to heat gradient-induced alignment?

We thank the reviewer for paying close attention to our supplementary videos. In that uniform heating experiment, the helix does indeed rotate, but it aligns at an angle to both helical \mathbf{q} -vectors. As there is no temperature gradient, we do not think this change is gradient-induced.

#4. In lines 45-48, the author claims their previous studies used a combination of external magnetic field and heat to transform skyrmion and reference Ref. 12-13. But it seems only the external field is applied. Or, by heat, do the authors mean controlling the temperature?

We thank the reviewer for mentioning this important detail. We do indeed mean controlling the temperature when we say “heat.” We have edited the mentioned text to clarify this and include the new text here for the reviewer’s convenience.

“A recent study has also found that externally applied magnetic field may drive the controlled, reversible topological transformations between magnetic Skys and Askys in Asky-hosting magnets^{12,13}”

#5. In previous work, the sample is Fe_{1.9}Ni_{0.9}Pd_{0.2}P. In this work, the authors use (Fe_{0.63}Ni_{0.3}Pd_{0.07})₃P. Are they the same?

We thank the reviewer for their attention to detail. Yes, Fe_{1.9}Ni_{0.9}Pd_{0.2}P is the same as (Fe_{0.63}Ni_{0.3}Pd_{0.07})₃P. We’ve recently switched to the latter notation to explicitly indicate that the compound is M₃P, where M is a transition metal.

#6. In line 57 and line 100 etc., the authors use the word ‘pins’. ‘Pins’ may indicate the external artifact-induced effect. Here, the Bloch-type chirality is preferred by the intrinsic DMI energy term. The author may want to change the word.

We thank the reviewer for this helpful comment. We have removed the word ‘pins’ and have replaced the text to read as follows.

“In (Fe_{0.63}Ni_{0.3}Pd_{0.07})₃P, anisotropic DMI stabilizes a ground state magnetic helix with two propagation vectors (\mathbf{q}) aligned almost along the [110] and [$\bar{1}$ 10] crystal axes¹²”

#7. In lines 135-136, the author states, 'suggests that Sky are more energetically favorable than Asky'. However, this is only valid under certain parameter ranges, as in this work, which is not universal. The author may want to modify the sentence to avoid confusion.

We thank the reviewer for their suggestion. We've modified the sentence in question and include the text here for the reviewer's convenience.

“Interestingly, we find that $\Delta E = E_{\text{Sky}} - E_{\text{Asky}} < 0$ in a magnetic polarized background for every M_S value considered here and so as T increases (decreasing M_S) a transformation from Asky towards Sky should be expected.”

#8. By the end of line 63, I guess the author wants to express “Even after the magnetic field is turned off...”, rather than “Once the magnetic field is turned off...”.

We thank the reviewer for their suggestion and have included the suggested wording in the modified main text.

#9. In the figure caption of Fig. 4, it shows that the temperature for the measurements in Fig. 4a is 295K, and Fig. 4c is 368K. What's the temperature for Fig. 4b? The procedure of applying heat gradients is a bit confusing.

We thank the reviewer for noticing this important detail. We have listed the temperature in the displayed field of view in the modified Fig. 3 caption (previously Fig. 4). Importantly, the temperature range in the field of view at the highest ∇T value (Fig. 3c, formerly Fig. 4c) is $343 \text{ K} < T_{\text{FOV}} < 368 \text{ K}$, and at this value, we didn't observe a significant decrease in antiskyrmion population. In contrast, the antiskyrmion population begins to decrease at $T = 330 \text{ K}$ during the uniform heating experiment (Fig. 4d, formerly Fig. 5d), suggesting that while the two experiments are qualitatively similar, they differ quantitatively, and the temperature gradient contributes to the antiskyrmions' robust metastability at zero field. We have added this discussion to the third paragraph in the section titled “ **∇T -driven topological transformation of Skys to Askys at zero field.**” We've included the modified text here for the reviewer's convenience.

“b-c, LTEM micrographs of the zero-field magnetic states taken after heater current applications of (b) $\nabla T = 2.0 \text{ K} \cdot \mu\text{m}^{-1}$ ($314 \text{ K} < T_{\text{FOV}} < 323 \text{ K}$, where T_{FOV} is the temperature in the field of view) and (c) $\nabla T = 5.2 \text{ K} \cdot \mu\text{m}^{-1}$ ($343 \text{ K} < T_{\text{FOV}} < 368 \text{ K}$) lasting 15 s (except for those values noted in the Methods).”

“In other words, ∇T is a sufficient, but not necessary condition for this topological transformation to occur. Importantly though, the temperature range in the field of view at the highest ∇T value (Fig. 3c) is $343 \text{ K} < T_{\text{FOV}} < 368 \text{ K}$, and at this value, we didn't observe a significant decrease in antiskyrmion population. In contrast, the antiskyrmion population begins to decrease at $T = 330 \text{ K}$ during the uniform heating experiment (Fig. 4d), suggesting that while the two experiments are qualitatively similar, they differ quantitatively, and the temperature gradient contributes to the antiskyrmions' robust metastability at zero field.”

Reviewer #3:

Heat current-driven transformation of antiskyrmions, non-topological bubbles and skyrmions while under a magnetic field and at zero field were observed by authors. The difference was found by them but the theoretical Interpretation was insufficient or say was lacked in the current state as was admitted by authors themselves as well. The experiment is complex to be honest, for example, the results shown in Figure 4 was taken after heater current applications, a cooling process was also occurred in this experiments I think. Therefore, I suggest a minor revision, and I suggest the paper was modified and resubmitted or was transfer to communication materials.

We thank the reviewer for their assessment that we only need a minor revision, and their recognition of the complexity of these in-situ experiments. We hope that the modifications we've made and described in detail above convince the reviewer of the merits of the current study.

In the Lorentz images of this manuscript, it is obvious that there is a contrast of black or white dots existing in skyrmions and antiskyrmions, which may be not consistent with conventional DMI skyrmions in FeGe or antiskyrmions in Mn1.4PtSn system, Can the authors explain this? Furthermore, the TIE analysis in Figure 1e,f and schematics of antiskyrmions and skyrmions may illustrate that there is not extra contrast in the center of these two objects as well, please confirm this!

We thank the reviewer for noticing this important detail. The reviewer is referring to the contrast located in the core of the spin textures. Indeed, sometimes this core contrast is present, and sometimes it isn't. We have been exploring this phenomenon in detail in a separate study, and we believe it relates to the stray field's effect on the three-dimensional structure of these spin textures. However, a three-dimensional treatment of these spin textures is beyond the scope of this study, which focuses on temperature and temperature gradient driven spin texture transformations.

We hope that our answers and in particular the many modifications we made in response to the other reviewers' comments are satisfactory to let the reviewer suggest publication in Nature Communications.

Reviewers' Comments:

Reviewer #1:

Remarks to the Author:

The authors did a great job in addressing my comments and the manuscript is now better organized, clearer, and improved. Still, I am not fully convinced about the key role of temperature gradients, at least for the transformation from skyrmions to antiskyrmions. However, I much appreciated the honesty and efforts of the authors in replying to my comments and, also, to a related comment from the second reviewer. I believe that discussing the differences between the two cases (uniform heating and thermal gradient) as well as giving a qualitative explanation has added clarity to the authors' results.

Despite my small remaining doubts, I will let the community debate about these achievements. Therefore, I recommend the revised version of the manuscript for publication in Nature Communication.

Reviewer #2:

Remarks to the Author:

The authors provide a detailed response to my previous questions and made lots of modifications, which have been nicely responded. And since the heat-related experiment technique are not straightforward to be controlled, and the authors present clearly that heat gradient's role on the new unique spin texture system, the manuscript present a well-designed experiment and nicely analysis study. Thus, I recommend this manuscript to be published in Nature communications.

I have two suggestions only for the authors' reference. The authors can decide by themselves to follow or not, which don't need my further review.

1. The difference in the energy term between S4 and D2d system might better be presented. Otherwise, some reader, who may not read the manuscript careful enough might have the impression that S4 and D2d systems are the same.
2. Since temperature could influence the bending contour. Thus, bending contour might be used as a calibration way that quantitatively shows the temperature as a function of time.

Response to Reviewer's Comments

First, we thank the reviewers for taking their time to review our manuscript, providing constructive comments for us to improve the quality of our work. We have carefully reviewed these comments and have incorporated them into the revised manuscript. Please find our point-by-point response to each review below. The original reviews are shown in *blue italics*. Additionally, we've **highlighted** all modifications within the revised manuscript.

Reviewer Comments:

Reviewer #1:

The authors did a great job in addressing my comments and the manuscript is now better organized, clearer, and improved. Still, I am not fully convinced about the key role of temperature gradients, at least for the transformation from skyrmions to antiskyrmions. However, I much appreciated the honesty and efforts of the authors in replying to my comments and, also, to a related comment from the second reviewer. I believe that discussing the differences between the two cases (uniform heating and thermal gradient) as well as giving a qualitative explanation has added clarity to the authors' results. Despite my small remaining doubts, I will let the community debate about these achievements. Therefore, I recommend the revised version of the manuscript for publication in Nature Communication.

We thank the reviewer for their comments and are pleased with their highly positive assessment of our response and recommendation to publish our manuscript in Nature Communications. We understand the reviewer's remaining doubts and appreciate their willingness to allow the physics community to discuss these results and perhaps add clarity with follow-up studies.

Reviewer #2:

The authors provide a detailed response to my previous questions and made lots of modifications, which have been nicely responded. And since the heat-related experiment technique are not straightforward to be controlled, and the authors present clearly that heat gradient's role on the new unique spin texture system, the manuscript present a well-designed experiment and nicely analysis study. Thus, I recommend this manuscript to be published in Nature communications.

We thank the reviewer for their comments and are pleased with their highly positive assessment of our manuscript and recommendation for publication in Nature Communications.

I have two suggestions only for the authors' reference. The authors can decide by themselves to follow or not, which don't need my further review.

1. The difference in the energy term between S4 and D2d system might better be presented. Otherwise, some reader, who may not read the manuscript careful enough might have the impression that S4 and D2d systems are the same.

We agree with the reviewer and have added a note in the supplementary text addressing this topic. We have included it here for the reviewer's convenience.

Note on helical q-vector orientation

The magnetic helices intrinsic to FNPP have modulation \mathbf{q} -vectors aligned at a finite angle to the $[110]$ and $[\bar{1}10]$ crystal axes. This contrasts with the helical \mathbf{q} -vectors in Heusler magnets with D_{2d} crystal symmetry, which are pinned to the $[100]$ and $[010]$ crystal axes. The specific pinning to these axes in D_{2d} stems from the additional rotational symmetries with respect to the $[100]$ and $[010]$ crystal axes. This manifests as an additional parameter in the DMI energy for S_4 systems whereas D_{2d} systems only have a single DMI parameter: ¹

$$H_{S_4} = -D_1(n_y \partial_x n_z - n_z \partial_x n_y) + D_1(n_z \partial_y n_x - n_x \partial_y n_z) - D_2(n_z \partial_x n_x - n_x \partial_x n_z) + D_2(n_z \partial_y n_y - n_y \partial_y n_z) \quad 1$$

and

$$H_{D_{2d}} = -D(n_y \partial_x n_z - n_z \partial_x n_y) + D(n_z \partial_y n_x - n_x \partial_y n_z), \quad 2$$

where D , D_1 , and D_2 are the DMI coefficients, and n_i is the i^{th} component of the normalized magnetization. In the D_1 terms of H_{S_4} and both $H_{D_{2d}}$ terms, the energy is minimized for Bloch twists of the magnetization along x (righthanded) and y (lefthanded), i.e., there is an energy penalty for twists that deviate from the yz -plane along the \hat{x} -direction or from the xz -plane along the \hat{y} -direction. The D_2 terms in H_{S_4} , however, favour Néel twists, i.e., twists within the xz -plane along the \hat{x} -direction or in the yz -plane along the \hat{y} -direction, clockwise along x and counterclockwise along y . If both D_1 and D_2 are nonzero, the DMI energy is minimized with Bloch twists along a \mathbf{q} -vector oriented at an angle away from x and y . As this angle can be clearly observed in Fig. 2a in the main text, we conclude that both D_1 and D_2 are nonzero in FNPP, where mostly D_2 is at work as it pins the helices in the $[110]$ and $[\bar{1}10]$ directions. Note, however, that in isotropic systems both energy functionals are equivalent up to a rotation of the coordinate system which we exploited for our simulations.

2. Since temperature could influence the bending contour. Thus, bending contour might be used as a calibration way that quantitatively shows the temperature as a function of time.

We thank the reviewer for this observation. Indeed, the change in diffraction contrast arising from the deformation of the thin plate as a temperature gradient is applied across it may be used to track temperature changes in real time. However, we've observed that this contrast is not entirely reliable as a quantitative measure, likely due to thin plate deformations that arise from the stress of excessive heating. It is likely possible to find a working temperature gradient range in which the stresses are sufficiently limited to enable such a quantitative measurement technique, and we leave it to future engineering studies to explore this further.